# A structure of substrate-bound Synaptojanin1 provides new insights in its mechanism and the effect of disease mutations

Jone Paesmans[1,2†], Ella Martin[1,2†], Babette Deckers[1,2], Marjolijn Berghmans[1,2], Ritika Sethi[1,2], Yannick Loeys[1,2], Els Pardon[1,2], Jan Steyaert[1,2], Patrik Verstreken[3,4], Christian Galicia[1,2*], Wim Versées[1,2*]

[1]VIB-VUB Center for Structural Biology, Brussels, Belgium; [2]Structural Biology Brussels, Vrije Universiteit Brussel, Brussels, Belgium; [3]VIB-KU Leuven Center for Brain and Disease Research, Leuven, Belgium; [4]KU Leuven, Department of Neurosciences, Leuven Brain Institute, Leuven, Belgium

**Abstract** Synaptojanin1 (Synj1) is a phosphoinositide phosphatase, important in clathrin uncoating during endocytosis of presynaptic vesicles. It was identified as a potential drug target for Alzheimer's disease, Down syndrome, and TBC1D24-associated epilepsy, while also loss-of-function mutations in Synj1 are associated with epilepsy and Parkinson's disease. Despite its involvement in a range of disorders, structural, and detailed mechanistic information regarding the enzyme is lacking. Here, we report the crystal structure of the 5-phosphatase domain of Synj1. Moreover, we also present a structure of this domain bound to the substrate diC8-PI(3,4,5)P$_3$, providing the first image of a 5-phosphatase with a trapped substrate in its active site. Together with an analysis of the contribution of the different inositide phosphate groups to catalysis, these structures provide new insights in the Synj1 mechanism. Finally, we analysed the effect of three clinical missense mutations (Y793C, R800C, Y849C) on catalysis, unveiling the molecular mechanisms underlying Synj1-associated disease.

\*For correspondence:
Christian.Galicia.Diaz.Santana@
vub.be (CG);
wim.versees@vub.be (WV)

[†]These authors contributed equally to this work

## Introduction

Phosphoinositides (PIPs) are membrane lipids that, together with their corresponding soluble inositol phosphates (IPs), regulate various cellular processes, including membrane recruitment of proteins, actin polymerization, synaptic vesicle trafficking and exo- and endocytosis (*Di Paolo and De Camilli, 2006*; *Balla, 2013*; *Ueda, 2014*). The dynamic control of the membrane distribution and relative abundance of the seven naturally occurring PIPs by a multitude of phosphoinositide kinases and phosphatases forms a versatile signaling mechanism able to tune the spatial and temporal regulation of many crucial events in the cell (*Fruman et al., 1998*; *Anderson et al., 1999*; *Hsu and Mao, 2015*).

The inositol polyphosphate 5-phosphatases (5PPases) form a family of $Mg^{2+}$-dependent enzymes, containing ten members in mammals, that catalyse the hydrolytic removal of the phosphate group on the 5-position of lipid-bound and soluble inositol phosphates (*Figure 1*, *Figure 1—figure supplement 1*). Based on their substrate specificity, the mammalian 5PPases can be further subdivided into four groups (*Majerus et al., 1999*): the type I 5PPase INPP5A (*Speed et al., 1996*), the type II 5PPases OCRL, INPP5B, INPP5J (or PIPP), SKIP, Synaptojanin1 (Synj1) and Synaptojanin 2 (Synj2) (*Lowe, 2005*; *Trésaugues et al., 2014*; *Mochizuki and Takenawa, 1999*; *Ijuin et al., 2000*; *McPherson et al., 1996*; *Nemoto et al., 1997*), the type III 5PPases, SHIP1 and SHIP2

**Figure 1.** Domain organization of the inositol polyphosphate 5-phosphatase family (5PPases). The domain organization of the ten human 5PPases, subdivided in four groups (type I–IV), is shown schematically. The different splice forms for Synaptojanin 1 (Synj1) and 2 (Synj2) are also shown. The domain boundaries of the 5PPase domain of Synj1 145 kDa used in this study and the disease mutations under study are indicated. 5PPase = 5-phosphatase domain; PH = Pleckstrin homology domain; ASH = ASPM, SPD-2, Hydin domain; Rho-GAP = Rho GTPase-activating protein domain; CB = clathrin-binding domain; PRD = proline-rich domain; SKICH = SKIP carboxyl homology domain; SAC1 = suppressor of actin 1-like domain; RRM = RNA recognition motif; SH2 = Src homology two domain; SAM = sterile alpha motif.

The online version of this article includes the following figure supplement(s) for figure 1:

**Figure supplement 1.** Sequence alignment of the 5PPase domains of ten human 5-phosphatases and *Schizosaccaromyces pombe* Synaptojanin (SPSynj).

(*Rohrschneider et al., 2000*; *Le Coq et al., 2017*), and the type IV 5PPase INPP5E (or Pharbin) (*Asano et al., 1999*). Among the type II 5PPases, the closely related Synj1 and Synj2 contain a similar domain arrangement, with in addition to the central 5PPase domain, an N-terminal suppressor of actin 1 (SAC1)-like domain and a C-terminal proline-rich domain (PRD). As such, Synj1 and Synj2 are unique in having two phosphatase activities, where the 5PPase domains can hydrolyse $PI(4,5)P_2$, $PI(3,4,5)P_3$, $IP_3$, and $IP_4$, while the SAC1-like domain can degrade $PI(3)P$, $PI(4)P$, and $PI(3,5)P_2$ (*Hsu and Mao, 2015*; *Cestra et al., 1999*; *Whisstock et al., 2002*). Both Synaptojanin proteins are implicated in clathrin-mediated endocytosis, where Synj2 is involved in the early stages of this process (*Rusk et al., 2003*), while the brain-specific 145 kDa splice isoform of Synj1 promotes clathrin uncoating during the late stages of endocytosis (*Perera et al., 2006*; *Ramjaun and McPherson, 1996*). In agreement with this role, knock-out studies of Synj1 in mice (*Cremona et al., 1999*) and *Drosophila melanogaster* (*Verstreken et al., 2003*) show endocytic defects at the neuronal synapses, indicating that Synj1 plays a critical role in synaptic function. This key function at the synapse is further illustrated by the implication of Synj1 in several diseases. Brain autopsy of Down syndrome (DS) patients revealed an excessive expression of the Synj1 protein, that is encoded on the triplicated chromosome 21, and it was shown that the overexpression of Synj1 leads to $PI(4,5)P_2$ deficiency and learning deficits in Down syndrome model mice (*Voronov et al., 2008*; *Arai et al.,*

*2002*). Elevated levels of Synj1 are also found in individuals showing a high risk for the development of Alzheimer's disease (AD) (*Berman et al., 2008*; *Martin et al., 2015*; *Miranda et al., 2018*). Based on these findings, Synj1 has been proposed as a potential attractive two-faced target for novel DS and AD treatments (*Cossec et al., 2012*). Additionally, inhibition of the 5-phosphatase activity of Synj1 has also been found to hold promise toward drug development for TBC1D24-associated epilepsy and DOORS syndrome (*Fischer et al., 2016*; *Lüthy et al., 2019*). On the other hand, recessive loss-of-function mutations in Synj1 are associated with either early-onset atypical parkinsonism or refractory epileptic seizures with severe progressive neurodegeneration (*Hardies et al., 2016*; *Xie et al., 2019*; *Taghavi et al., 2018*; *Hong et al., 2019*; *Bouhouche et al., 2017*).

Out of the ten 5PPase domains present in human, high-resolution structural information has only been published for three representatives: INPP5B, OCRL, and SHIP2 (*Trésaugues et al., 2014*; *Mills et al., 2016*). In addition, an unpublished structure of INPP5E has been deposited in the protein data bank (PDB 2XSW). So far no experimentally determined structure is available for the 5PPase domain of either Synj1 or Synj2, although the very first 5PPase structure ever to be determined was the one of a Synaptojanin homolog of the yeast *Schizosaccharomyces pombe* (*Tsujishita et al., 2001*).

In addition to the structural data of the 5PPase domains in their apo form, crystal structures in complex with reaction products and inhibitors, together with detailed studies on the structurally and mechanistically related Apurinic/Apyrimidinic endonucleases (APE), have yielded some insights in the catalytic mechanism of the hydrolysis reaction (*Whisstock et al., 2000*). Indeed, structural and sequence comparison revealed similarities in the active site architecture of the 5PPases and APE1, a $Mg^{2+}$-dependent enzyme that catalyses the cleavage of the phosphodiester bond on the 5' side of the abasic site in DNA using a conserved aspartate as catalytic base and a conserved histidine that presumably stabilizes the phosphorane transition state. Moreover, leaving group activation has been proposed to occur via a $Mg^{2+}$-bound water molecule, although the exact role of the $Mg^{2+}$-ion in the catalytic mechanism of 5PPases has not been fully established (*Trésaugues et al., 2014*; *Mills et al., 2016*; *Aboelnga and Wetmore, 2019*). A crystal structure of the *S. pombe* Synaptojanin (SPSynj) homolog in complex with inositol-(1,4)-bisphosphate provided a first snapshot of a potential enzyme-product complex (*Tsujishita et al., 2001*). However, subsequent structures of the catalytic domain of INPP5B in complex with diC8-PI(4)P and diC8-PI(3,4)$P_2$ revealed another orientation of these reaction products, suggesting that the placement of the ligand in the SPSynj structure does not correspond to the genuine position of the product (*Trésaugues et al., 2014*). Further elucidation of the complete hydrolysis mechanism is however hampered by the lack of 5PPase structures in complex with reaction substrates showing directly the exact placement of the 5-phosphate group in the active site.

In this study we report the first structure of the catalytic 5PPase domain of human Synj1 by using Nanobody-aided crystallography. Moreover, we were able, for the first time, to solve a structure of a 5PPase domain with the substrate diC8-PI(3,4,5)$P_3$ trapped in its active site, revealing the placement of, and the interactions with, the 5-phosphate group. In combination with a detailed kinetic analysis of the hydrolysis reaction catalysed by the Synj1 5PPase domain, these structures provide additional insights in the catalytic mechanism. Finally, we investigated in detail the effect of the three currently described homozygous missense mutations in the 5PPase domain (Y793C, R800C, Y849C) associated with either (young onset) Parkinson's disease or intractable epilepsy with neurodegeneration (*Hardies et al., 2016*; *Xie et al., 2019*; *Taghavi et al., 2018*) on the reaction rate for different substrates, thus providing insights in the molecular mechanisms underlying these diseases.

## Results

### The crystal structure of the 5-phosphatase domain of Synj1 and its complex with the substrate diC8-PI(3,4,5)$P_3$

Since structural information regarding the 5-phosphatase (5PPase) domain of human Synj1 is currently lacking, we set-out to crystallize a construct of the 5PPase domain spanning residues 528–873 (Synj1$_{528–873}$). Upon multiple unsuccessful attempts to obtain well diffracting crystals, we turned to Nanobody (Nb)-assisted crystallization. After llama immunization, library construction and two successive rounds of phage display panning, six Synj1$_{528–873}$-specific Nb families were obtained (data

not shown). Various Synj1$_{528-873}$-Nb complexes were used to set-up crystallization screens and well diffracting crystals belonging to space group C121 were obtained for the complex between Synj1$_{528-873}$ and Nb13015 (hereafter called Nb15).

The first dataset was collected and refined at 2.3 Å resolution (*Table 1*). The structure was solved using molecular replacement, revealing three Nb15-Synj1$_{528-873}$ complexes in each asymmetric unit (AU) (*Figure 2—figure supplement 1A*). Since Synj1$_{528-873}$ consistently behaves as a monomer in solution, as assessed through gel filtration, it can be assumed that the three complexes in the AU are merely interacting via crystallographic contacts. Despite being soaked overnight with IP$_6$ (inositol-1,2,3,4,5,6-hexakisphosphate), no density accounting for this molecule was observed in the active site, and therefore we will further refer to this structure as an apo-structure. However, several blobs of density in the structure could be modelled as orthophosphates, suggesting that these result from hydrolysis of IP$_6$ (*Figure 2—figure supplement 2*). Furthermore, all three Synj1$_{528-873}$ molecules contain density close to N543 and E591, which corresponds to the position of a Mg$^{2+}$-ion in other 5PPase structures (*Trésaugues et al., 2014*), and which was therefore also here modelled as Mg$^{2+}$-ions (*Figure 2—figure supplement 2*). While these Mg$^{2+}$-ions were modelled at a very similar position as in the structure of INPP5B in complex with the product diC8-PI(3,4)P$_2$ (*Trésaugues et al., 2014*), it must be noted that the distances to residues N543 and E591 are rather long.

Superposition of the three Synj1$_{528-873}$ molecules present in the AU shows that they are very similar, with root-mean square deviations (rmsd) for superposition of all main chain atoms of chain A on chain C and E of 0.69 Å and 0.49 Å, respectively (*Figure 2—figure supplement 1B*). Overall, Synj1$_{528-873}$ adopts a fold that is very similar to the fold of the catalytic domain of other 5-phosphatases (*Figure 2—figure supplement 3*, *Supplementary file 1A*), and is composed of two β sheets forming a β-sandwich surrounded by seven α-helices (*Figure 2A*; *Trésaugues et al., 2014*; *Tsujishita et al., 2001*). Nb15 forms interactions via its three CDR loops with a loop and short 3$_{10}$-helix (aa 641–654) connecting β4 and β5 of Synj1$_{528-873}$, on the opposite side of the 5PPase active site, thus leaving the active site open for ligand binding (*Figure 2A*).

A second crystal of the Nb15-Synj1$_{528-873}$ complex, was soaked with 1 mM diC8-PI(3,4,5)P$_3$ and data were collected and refined at 2.73 Å resolution (*Table 1*). Analysis of the electron density in the active sites of the three molecules in the asymmetric unit revealed that one active site (corresponding to chain A) contains unambiguous density for the diC8-PI(3,4,5)P$_3$ substrate including the scissile 5-phosphate group (*Figure 2B–C*, *Figure 2—figure supplement 1C*, *Figure 3*), thus showing that we were able to trap the non-hydrolysed substrate by using a strategy of short substrate soaking followed by flash freezing and assisted by the slower substrate turnover under the conditions used for crystallization (*Figure 4—figure supplement 1*, *Figure 4—figure supplement 1—source data 1*). The other two Synj1$_{528-873}$ active sites (corresponding to chain C and E) contain weaker and discontinuous electron density, probably due to substrate already being hydrolysed to a larger extent because of subtle differences in kinetics of substrate access/hydrolysis and product release in these sites depending on the local environment of the crystal packing. The electron density in these active sites was modelled as phosphate ions, with three phosphates in chain C at positions corresponding to the 1-, 4-, and 5-phosphates of diC8-PI(3,4,5)P$_3$, while in chain E only a single phosphate could be modelled between the expected position of the phosphates present on the 4- and 5-position of the substrate (*Figure 2—figure supplement 2*). Analysis of the density revealed that two out of three Synj1$_{528-873}$ molecules (chain A and C) also show density close to residues N543 and E591, where Mg$^{2+}$-ions were modelled (*Figure 2—figure supplement 2*). Despite the difference in the occupancy of diC8-PI(3,4,5)P$_3$ in the active sites, superposition of the Synj1$_{528-873}$ molecules (corresponding to chain A, C, and E) does not reveal any large differences in active site loops and residues, indicating that no significant substrate-induced conformational changes take place (*Figure 2D*).

## Enzyme-substrate interactions in the Synj1$_{528-873}$– diC8-PI(3,4,5)P$_3$ complex

The current crystal structure of Synj1$_{528-873}$ bound to diC8-PI(3,4,5)P$_3$ provides the first experimental structural view of any inositol polyphosphate 5-phosphatase (5PPase) in complex with a trapped genuine substrate, allowing us to describe and analyse the enzyme-substrate interactions in detail.

In particular, the Synj1$_{528-873}$-diC8-PI(3,4,5)P$_3$ structure for the first time reveals the exact location and interactions with the scissile phosphate (5 P). This phosphate is oriented towards two regions previously described to correspond to conserved sequence motifs characteristic for the 5PPase

**Table 1.** Data collection and refinement statistics.

| | Synj1$_{528-873}$ - Apo | Synj1$_{528-873}$ - diC8-PI(3,4,5)P$_3$ |
|---|---|---|
| PDB code | 7A0V | 7A17 |
| **Data collection** | | |
| Synchrotron | Diamond | Soleil |
| Beamline | i03 | Px2a |
| Wavelength (Å) | 0.98 | 0.98 |
| Resolution range (Å)* | 87.06–2.30 (2.43–2.30) | 87.39–2.73 (3.02–2.73) |
| Space group | C121 | C121 |
| Unit cell dimensions (Å) | a = 168.87 | a = 169.32 |
| | b = 108.79 | b = 109.21 |
| | c = 100.97 | c = 100.90 |
| Unit cell angles (°) | α = 90.00 | α = 90.00 |
| | β = 120.72 | β = 120.62 |
| | γ = 90.00 | γ = 90.00 |
| Spherical completeness (%)* | 77.1 (26.4) | 76.2 (22.8) |
| Ellipsoidal completeness (%)* | 92.3 (95.0) | 91.5 (57.3) |
| Unique reflections | 53789 | 32149 |
| Mean (I)/SD(I)* | 11.2 (1.4) | 5.3 (1.4) |
| CC(1/2)* | 0.997 (0.512) | 0.964 (0.474) |
| Multiplicity* | 7.0 (5.7) | 3.5 (3.6) |
| R$_{meas}$ (%)* | 14.5 (125.1) | 28.2 (123.7) |
| **Refinement** | | |
| Resolution range (Å) | 86.81–2.30 | 86.83–2.73 |
| R$_{work}$ (%) | 19.64 | 19.88 |
| R$_{free}$ (%)† | 25.23 | 25.74 |
| Model content | | |
| Molecules per AU | 6 | 6 |
| Protein atoms per AU | 10574 | 10624 |
| Ligand atoms per AU | 39 | 85 |
| Metal atoms per AU | 3 | 2 |
| Water molecules per AU | 359 | 71 |
| Wilson B factors (Å$^2$) | 38.08 | 43.28 |
| Average B factors (Å$^2$) | | |
| Protein atoms | 47.35 | 44.88 |
| Ligand atoms | 65.62 | 67.11 |
| Metal atoms | 60.08 | 42.62 |
| Water molecules | 42.68 | 22.93 |
| Rmsd bonds (Å) | 0.002 | 0.006 |
| Rmsd angles (°) | 0.456 | 1.15 |
| Ramachandran plot (%) (favored, outliers) | 95.79, 0.23 | 97.00, 0.46 |

* Values in parentheses are for the high-resolution shell.

† R$_{free}$ is based on a subset of 5% of reflections omitted during refinement.

AU, asymmetric unit.

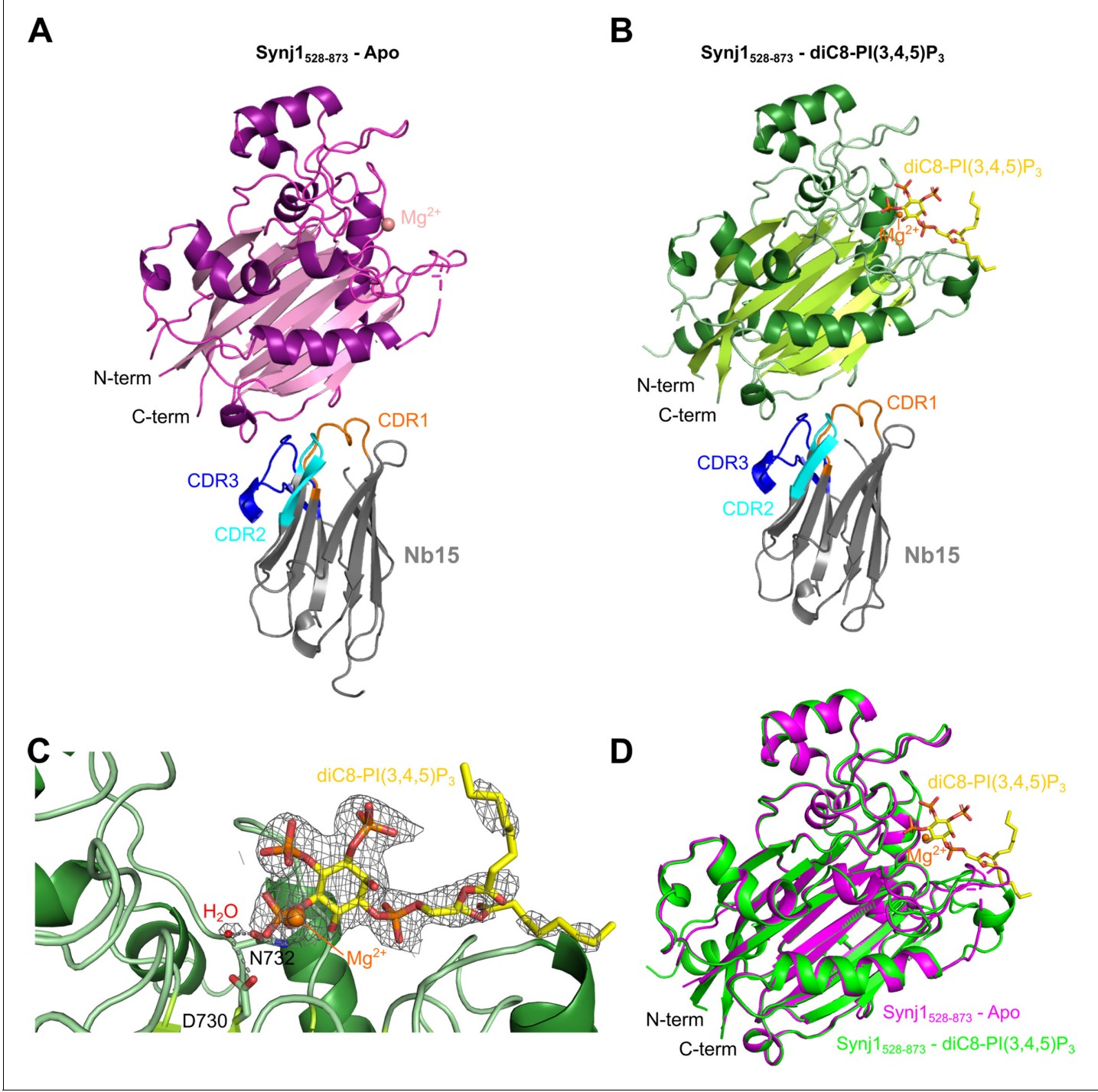

**Figure 2.** Structure of the Nb15-Synj1$_{528–873}$ complex in presence or absence of the substrate diC8-PI(3,4,5)P$_3$. (A) Apo-structure of the Nb15-Synj1$_{528–873}$ complex. Synj1$_{528–873}$ (chain E) is represented in different shades of magenta, while the Nb (chain F) is represented in grey with indication of the different CDR regions. The Mg$^{2+}$-ion is shown as a salmon sphere. (B) The Nb15-Synj1$_{528–873}$ complex bound to diC8-PI(3,4,5)P$_3$. Synj1$_{528–873}$ (chain A) is represented in different shades of green, while the Nb (chain B) is represented similar as in (A). The Mg$^{2+}$-ion is shown as an orange sphere and diC8-PI(3,4,5)P$_3$ is shown as yellow sticks. (C) Zoom-in on the active site region of Synj1$_{528–873}$ with bound diC8-PI(3,4,5)P$_3$ (yellow sticks), Mg$^{2+}$ (orange sphere) and the nucleophilic water molecule (red sphere) shown with their corresponding 2F$_O$-F$_C$-map contoured at 1σ. Residues D730 and N732, which play a role in the activation of the nucleophilic water, are shown as green sticks. (D) Superposition of the apo (magenta) and diC8-PI(3,4,5)P$_3$-bound (green) Synj1$_{528–873}$ structure.

The online version of this article includes the following figure supplement(s) for figure 2:

**Figure supplement 1.** Content of the asymmetric unit (AU) of the apo and the diC8-PI(3,4,5)P$_3$-bound Nb15-Synj1$_{528–873}$ complex.

*Figure 2 continued on next page*

*Figure 2 continued*

**Figure supplement 2.** Close-up view on the active site of different Synj1$_{528-873}$-chains of the apo- and diC8-PI(3,4,5)P$_3$-bound Nb15-Synj1$_{528-873}$ structure.

**Figure supplement 3.** Superposition of the catalytic (5PPase) domain of human Synj1$_{528-873}$ on the available structures of the other human 5-phosphatases and SPSynj.

family: WXG**D**X**N**(Y/F)**R** (residues 727–734) and P(A/S)W(C/T)**D**R(I/V)L (residues 802–809) (*Jefferson and Majerus, 1996*), while it also interacts with a number of other highly conserved active site residues (*Figure 3*, *Figure 1—figure supplement 1*, *Figure 2—figure supplement 3*, *Supplementary file 1B*). The first oxygen of 5 P (O$_{PH}$) is within hydrogen bonding distance of N732 (3.0 Å), while it is also appropriately oriented to form a weak hydrogen bond or salt bridge with H689 (3.6 Å). The second oxygen of 5 P (O$_{PF}$) can form a (weak) salt bridge or hydrogen bond with H859 (3.3 Å). The O$_{PF}$ oxygen is also oriented towards the Mg$^{2+}$-ion, although, in contrast to what was previously suggested (*Trésaugues et al., 2014*), the Mg$^{2+}$-O$_{PF}$ distance of 4.7 Å is too long to account for a direct metal coordination interaction. The third oxygen of 5 P (O$_{PG}$) potentially forms a weak hydrogen bond with Y784 (3.5 Å) and an electrostatic interaction with R734 (3.5 Å) (*Figure 3*). D730 and N732 form a pair of conserved residues belonging to one of the conserved sequence motifs of the 5PPase family. It has previously been shown that D730 corresponds to the general base required to activate a water molecule for nucleophilic attack on P$_5$. Analysis of the electron density shows such a water molecule, located at 3.1 Å from D730 and 3.5 Å from the P$_5$-atom (*Figure 2C* and *3*). Additionally, the structure also reveals a hydrogen bond between the water molecule and N732 (2.6 Å), probably required for proper orientation of the water molecule for nucleophilic attack. This thus shows a direct role of the D730/N732 pair in activation of the nucleophilic water.

The phosphate on position 4 (4 P) forms extensive interactions with the enzyme through the conserved P4-interacting-motif (P4IM), containing Y784, K798, and R800 (*Figure 3*; *Figure 1—figure supplement 1*; *Trésaugues et al., 2014*; *Mills et al., 2012*). These multiple binding interactions explain the preference of Synj1$_{528-873}$ and most other 5PPases for substrates phosphorylated on the 4-position (*Schmid et al., 2004*). In addition, the Mg$^{2+}$-ion is located at a distance of 3.6 Å from the O$_{9P}$-atom of the 4 P group. It is also noteworthy that a non-proline *cis* peptide bond is found between active site residues Y784 and K785, belonging to the P4IM region. This *cis* peptide bond is also conserved in the structure of OCRL, INPP5B, SHIP2, and INPP5E, and, while it was not modelled as such in the SPSynj structure, the density can account for a *cis* peptide in that latter structure. While the occurrence of such bonds is mostly of functional importance (*Jabs et al., 1999*), the exact relevance to the 5PPase mechanism is not entirely clear as it is found both in the apo and ligand-bound structures.

In contrast to the 4 P and 5 P groups, the phosphate on position 3 (3 P) is solvent exposed and does not form any interaction with enzyme residues. The only direct contacts between Synj1$_{528-873}$ and the inositol group of diC8-PI(3,4,5)P$_3$ are mediated by a weak Van der Waals interaction (3.8 Å) between the β-carbon of A692 and the C$_5$ atom of the inositol ring (*Figure 3*). Additionally, K669 is located at 2.8 Å from the OH-group on position 6 of the inositol ring, forming a hydrogen bond. Besides the inositol ring, also the phosphate on position 1 (1 P) is commonly used by all 5PPases for substrate recognition and binding (*Figure 2—figure supplement 3*, *Supplementary file 1B*). In our structure, the 1 P of diC8-PI(3,4,5)P$_3$ forms strong interactions with the side chains of N668 (2.2 Å) and K669 (2.7 Å) (*Figure 3*). Finally, the lipid anchors of diC8-PI(3,4,5)P$_3$ are interacting with two hydrophobic regions, as also described for other 5PPases (*Trésaugues et al., 2014*; *Mills et al., 2012*). The first region is formed by residues V593 to T606 and has been called lipid chain 1 recognition motif (LC1R), while the second region is formed by residues T660 to N668 and has been called lipid chain 2 recognition motif (LC2R) (*Figure 1—figure supplement 1*; *Trésaugues et al., 2014*).

## Kinetic analysis of the Synj1 5-phosphatase activity

A detailed and systematic kinetic analysis of the contribution of the different phosphate groups of the phosphoinositide substrate to Synj1 catalysis is currently lacking. To investigate the contribution of the different inositol phosphate groups (3 P, 4 P, and 5 P) and the Mg$^{2+}$-cofactor on binding and

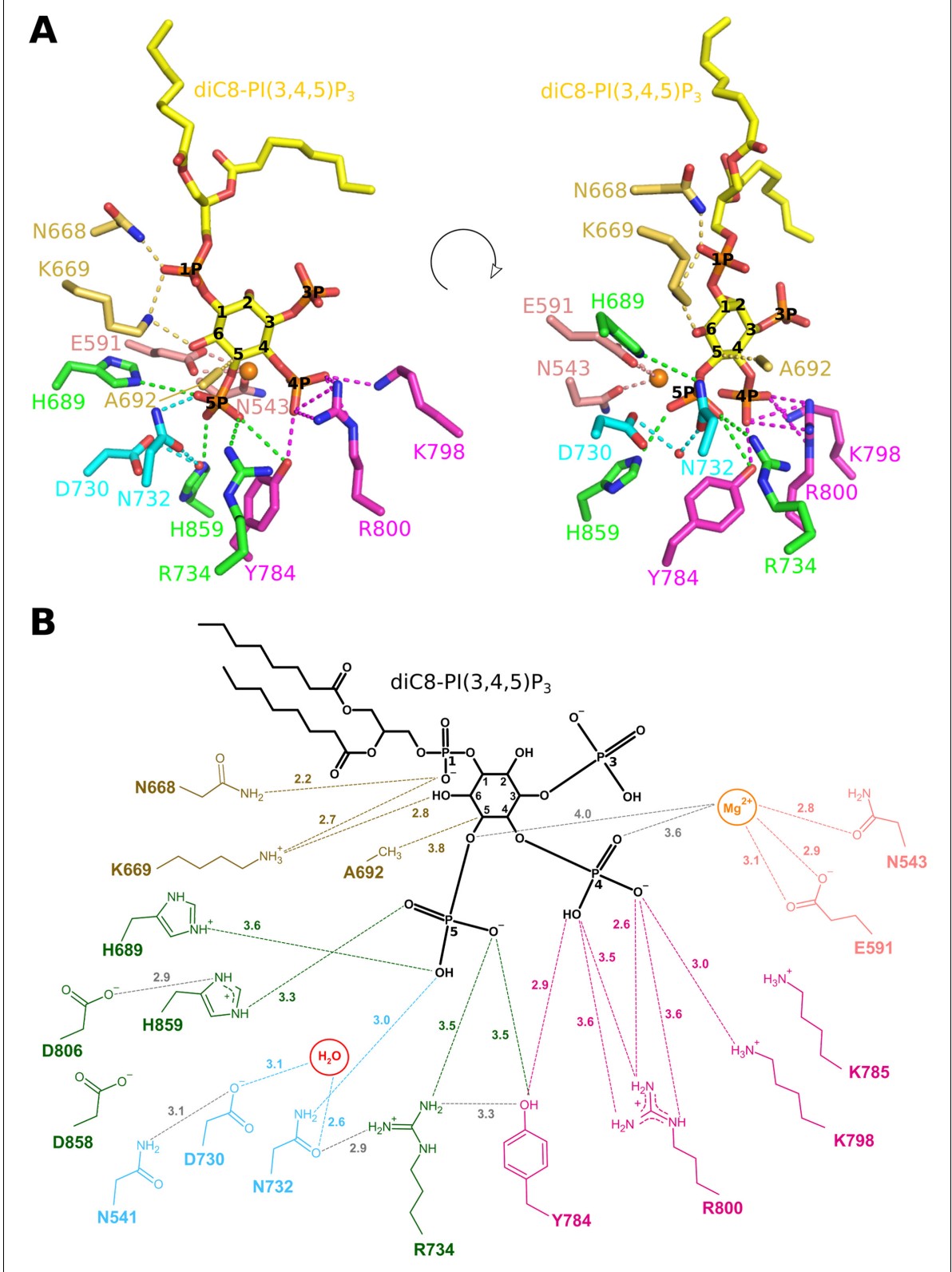

**Figure 3.** Enzyme-substrate interactions in the Synj1$_{528-873}$-diC8-PI(3,4,5)P$_3$ complex. (**A**) Zoom-in on the active site where Synj1$_{528-873}$ forms interactions with different groups of the diC8-PI(3,4,5)P$_3$ substrate (yellow sticks). The residues coloured in gold are forming interactions with the 1 P group or inositol ring of the PIP (gold dashes), residues coloured in magenta form interactions with the 4 P group (magenta dashes), and residues shown in green are interacting with the 5 P group (green dashes). Residues important for activation of the nucleophilic water (red sphere) are shown in cyan,

*Figure 3 continued on next page*

*Figure 3 continued*

while two residues shown in salmon are interacting with the $Mg^{2+}$-ion (orange sphere). (B) Schematic representation of the interactions between $Synj1_{528-873}$ and diC8-PI(3,4,5)$P_3$. The same colour-code as in (A) was used, except for the substrate that is shown in black. Distances (in Å) between interacting atoms are indicated.

substrate turnover, we therefore performed a full steady-state kinetic analysis of $Synj1_{528-873}$, using $IP_3$, diC8-PI(3,4,5)$P_3$, diC8-PI(4,5)$P_2$, diC8-PI(3,5)$P_2$, and diC8-PI(5)P as substrates (*Figure 4A*, *Table 2*, *Figure 4—source data 1*).

Based on the overall specificity constant ($k_{cat}/K_M$) the following order in substrate preference is observed: diC8-PI(4,5)$P_2$ ≈ diC8-PI(3,4,5)$P_3$ > $IP_3$ >> diC8-PI(5)P ≈ diC8-PI(3,5)$P_2$. While this trend is similar to what has been previously reported based on activity measurements at a single substrate concentration (*Schmid et al., 2004*), it differs from the substrate preference profile of SPSynj where the following profile was found: $IP_3$ ≈ diC4-PI(4,5)$P_2$ > diC4-PI(3,5)$P_2$ ≈ diC4-PI(3,4,5)$P_3$ (*Chi et al., 2004*). This indicates that the yeast homolog is not an ideal model system to study the mechanism of human Synj1.

The contribution of the acyl chains to catalysis can be deduced by comparing the kinetic parameters of diC8-PI(4,5)$P_2$ and its corresponding head group $IP_3$ (*Figure 4B*). This comparison shows that diC8-PI(4,5)$P_2$ has a nearly 13-fold higher $k_{cat}/K_M$ value than $IP_3$, due to a 2.5-fold higher affinity (lower $K_M$), but mainly due to the 5-fold higher turnover rate ($k_{cat}$), suggesting that the acyl chains are required to properly orient the head group in the active site for catalysis.

The contribution of the 4 P group to substrate binding and turnover can be obtained by comparing either diC8-PI(3,4,5)$P_3$ with diC8-PI(3,5)$P_2$ or diC8-PI(4,5)$P_2$ with diC8-PI(5)P (*Figure 4B*). Overall, this analysis for the $k_{cat}/K_M$ value reveals a very large contribution of the 4 P group to Synj1 activity by more than 3 orders of magnitude (corresponding to $\Delta\Delta G_{overall}$ = 4.3–4.7 kcal/mol). Interestingly, comparing the individual kinetic constants ($k_{cat}$ and $K_M$) of diC8-PI(3,4,5)$P_3$ and diC8-PI(3,5)$P_2$ reveals that this overall contribution is mainly attributed to the catalytic turnover ($k_{cat}$), with the 4 P group contributing a factor 340 to catalysis (corresponding to $\Delta\Delta G_{catalysis}$ = 3.5 kcal/mol). In contrast, the 4 P group only contributes relatively little to ligand binding, with removal of the 4 P group leading to a 7-fold increase in the $K_M$ value (corresponding to $\Delta\Delta G_{binding}$ = 1.2 kcal/mol). To the best of our knowledge, this is the first time it is unequivocally shown that the phosphoinositide 4 P group directly contributes to substrate turnover by the Synj1 5PPase domain, having important consequences for its catalytic mechanism (see Discussion).

On the other hand, comparing diC8-PI(3,4,5)$P_3$ with diC8-PI(4,5)$P_2$ shows an overall small effect of the 3 P group on the catalytic parameters, which corresponds to the solvent-exposed 3 P group in our structure.

Finally, the contribution of the $Mg^{2+}$-ion to catalysis was quantified by comparing the catalytic parameters for diC8-PI(4,5)$P_2$ between $Mg^{2+}$-bound and $Mg^{2+}$-free $Synj1_{528-873}$ (*Figure 4B*). Removal of $Mg^{2+}$ severely impacted activity with a decrease in the specificity constant ($k_{cat}/K_M$) by 4 orders of magnitude ($\Delta\Delta G_{overall}$ = 5.4 kcal/mol). This decrease in activity is mainly caused by a decrease in substrate turnover (1400-fold decrease in $k_{cat}$, $\Delta\Delta G$ = 4.3 kcal/mol) rather than an effect on substrate binding (7.5-fold increase in $K_M$, $\Delta\Delta G_{binding}$ = 1.2 kcal/mol). The observed contribution of the $Mg^{2+}$-ion mainly to catalysis rather than substrate binding is in good agreement with the structure of the enzyme-substrate-complex. In this structure, the $Mg^{2+}$-ion is located at a relatively large distance from the substrate's 4 P group (3.6 Å), accounting for a relatively weak binding interaction. On the other hand, its contribution to catalysis can be accounted for by either a role in water-mediated leaving group activation and/or by a stabilizing interaction with the phosphorane transition state (see Discussion).

## Impact of missense disease mutations on the Synj1 5-phosphatase activity

Missense and nonsense mutations in the 5PPase domain of Synj1 have been associated with several neurological disorders, such as early-onset seizures and early-onset atypical Parkinson's disease (*Hardies et al., 2016*; *Xie et al., 2019*; *Taghavi et al., 2018*; *Hong et al., 2019*; *Bouhouche et al., 2017*). At the moment of writing this manuscript, three homozygous point variants in the 5PPase domain of Synj1 had been described in patients: the Y793C mutation leading to typical levodopa-

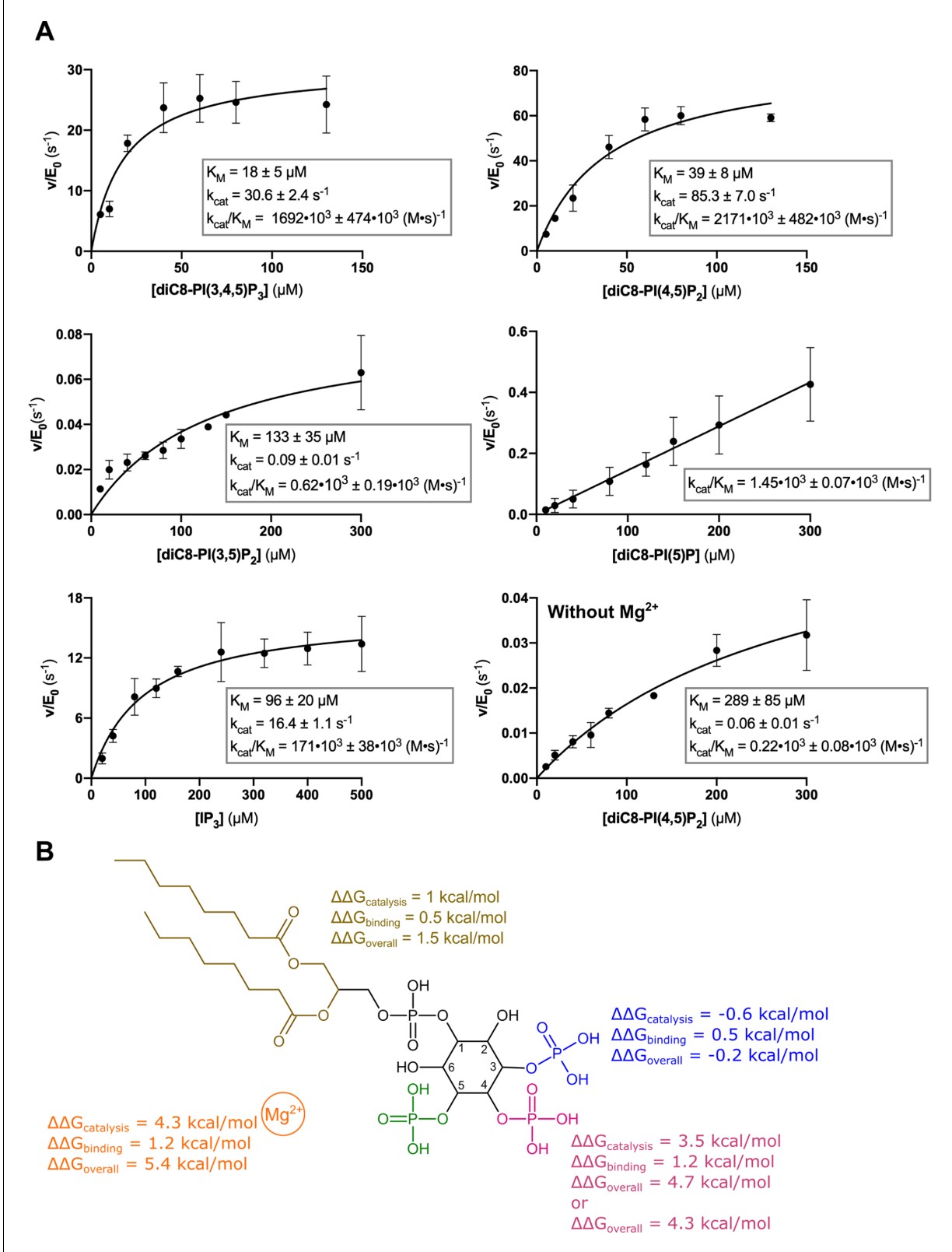

**Figure 4.** Kinetic analysis of the contribution of different substrate groups to the Synj1$_{528-873}$ 5-phosphatase activity. (A) Michaelis-Menten curves obtained for Synj1$_{528-873}$ with different substrates: diC8-PI(3,4,5)P$_3$, diC8-PI(4,5)P$_2$, diC8-PI(3,5)P$_2$, diC8-PI(5)P, IP$_3$, and diC8-PI(4,5)P$_2$ in the absence of Mg$^{2+}$. The turnover number (k$_{cat}$), the Michaelis-Menten constant (K$_M$) and specificity constant (k$_{cat}$/K$_M$) are given for every measurement, together with the standard error. Each datapoint is the average of three independent measurements with the error bars representing the standard deviation. (B)
*Figure 4 continued on next page*

*Figure 4 continued*

Schematic overview of the contribution of the different groups of the diC8-PI(3,4,5)P$_3$-substrate to the Synj1$_{528-873}$ mechanism, with the acyl chains coloured in gold, the 3 P group in blue, the 4 P group in magenta, the 5 P group in green and the Mg$^{2+}$-ion in orange. The $\Delta\Delta G$ value shows the contribution of the acyl chains, 3 P, 4 P, and Mg$^{2+}$-ion to catalysis (k$_{cat}$), binding (K$_M$) and overall catalytic efficiency (k$_{cat}$/K$_M$). The more positive the $\Delta\Delta G$ value, the larger the contribution of the respective group to either catalysis, binding or overall catalytic efficiency.

The online version of this article includes the following source data and figure supplement(s) for figure 4:

**Source data 1.** Steady-state enzyme kinetics data of Synj1$_{528-873}$ wild-type in combination with different substrates.

**Figure supplement 1.** The 5-phosphatase activity of Synj1$_{528-873}$ is affected by Nb15 and acidic pH.

**Figure supplement 1—source data 1.** Steady-state enzyme kinetics data of Synj1$_{528-873}$ wild-type using diC8-PI(3,4,5)P$_3$ as substrate at pH 5.5 and in presence of an excess of Nb15 (comparable to the crystallization conditions).

**Figure supplement 2.** Steady-state enzyme kinetics of the Synj1$_{528-873}$ Y793C mutant in combination with different substrates.

**Figure supplement 2—source data 1.** Steady-state enzyme kinetics data of the Synj1$_{528-873}$ Y793C mutant in combination with different substrates.

**Figure supplement 3.** Steady-state enzyme kinetics of the Synj1$_{528-873}$ R800C mutant in combination with different substrates.

**Figure supplement 3—source data 1.** Steady-state enzyme kinetics data of the Synj1$_{528-873}$ R800C mutant in combination with different substrates.

**Figure supplement 4.** The (GST-tagged) Synj1$_{528-873}$ Y849C mutant shows no 5-phosphatase activity.

responsive parkinsonism (*Xie et al., 2019*), the R800C mutation leading to asymmetric parkinsonism and seizures (*Taghavi et al., 2018*), and the Y849C mutation leading to early-onset treatment-resistant seizures and progressive neurological decline (numbering based on Synaptojanin1-145 isoform 2) (*Hardies et al., 2016*; *Figure 1*). To aid in rationalizing the contribution of these mutations

**Table 2.** Steady-state kinetic parameters of Synj1$_{528-873}$ and the Synj1$_{528-873}$ Y793C, R800C and Y849C mutants in combination with different substrates.

| | | Synj1$_{528-873}$ | Synj1$_{528-873}$ Y793C | Synj1$_{528-873}$ R800C | Synj1$_{528-873}$ Y849C |
|---|---|---|---|---|---|
| diC8-PI(3,4,5)P$_3$ | k$_{cat}$ (s$^{-1}$) | 30.6 ± 2.4 | 7.6 ± 0.9 | 21.8 ± 4.2 | NMA |
| | K$_M$ (µM) | 18 ± 5 | 29 ± 10 | 155 ± 49 | NMA |
| | k$_{cat}$/K$_M$ (•10$^3$ (M•s)$^{-1}$) | 1692 ± 474 | 259 ± 97 | 141 ± 52 | NMA |
| diC8-PI(4,5)P$_2$ | k$_{cat}$ (s$^{-1}$) | 85.3 ± 7.0 | 32.0 ± 3.2 | 6.4 ± 1.2 | NMA |
| | K$_M$ (µM) | 39 ± 8 | 117 ± 25 | 161 ± 52 | NMA |
| | k$_{cat}$/K$_M$ (•10$^3$ (M•s)$^{-1}$) | 2171 ± 482 | 274 ± 64 | 40 ± 15 | NMA |
| IP$_3$ | k$_{cat}$ (s$^{-1}$) | 16.4 ± 1.1 | 1.5 ± 0.2 | 0.055 ± 0.007 | NMA |
| | K$_M$ (µM) | 96 ± 20 | 864 ± 171 | 289 ± 78 | NMA |
| | k$_{cat}$/K$_M$ (•10$^3$ (M•s)$^{-1}$) | 171 ± 38 | 1.7 ± 0.4 | 0.19 ± 0.06 | NMA |
| diC8-PI(3,5)P$_2$ | k$_{cat}$ (s$^{-1}$) | 0.09 ± 0.01 | 0.012 ± 0.001 | ND | ND |
| | K$_M$ (µM) | 133 ± 35 | 101 ± 25 | ND | ND |
| | k$_{cat}$/K$_M$ (•10$^3$ (M•s)$^{-1}$) | 0.62 ± 0.19 | 0.12 ± 0.03 | 0.29 ± 0.01 | ND |
| diC8-PI(5)P | k$_{cat}$ (s$^{-1}$) | ND | ND | ND | ND |
| | K$_M$ (µM) | ND | ND | ND | ND |
| | k$_{cat}$/K$_M$ (•10$^3$ (M•s)$^{-1}$) | 1.45 ± 0.07 | 0.14 ± 0.01 | 1.2 ± 0.1 | ND |
| diC8-PI(4,5)P$_2$ without Mg$^{2+}$ | k$_{cat}$ (s$^{-1}$) | 0.06 ± 0.01 | ND | ND | ND |
| | K$_M$ (µM) | 289 ± 85 | ND | ND | ND |
| | k$_{cat}$/K$_M$ (•10$^3$ (M•s)$^{-1}$) | 0.22 ± 0.08 | ND | ND | ND |

ND, not determined.

NMA, no measurable activity.

in the onset of disease, their impact on the kinetic parameters for different substrates was determined (**Table 2**).

The Y793 residue is present on the large loop that contains the P4IM. Although its side chain is pointing away from the active site, it is located in between active site residues Y784 and K785 on the one hand and K798 and R800 on the other hand, all located in the P4IM (**Figure 5A–B**). Y793 could potentially stabilize the conformation of this loop by making hydrogen bonds with Y786 and the main chain carbonyl of P782. The Parkinson's disease mutation Y793C has a clear but rather moderate effect on the Synj1$_{528–873}$ 5PPase activity. The largest decrease in activity is observed for IP$_3$, where the measured catalytic efficiency ($k_{cat}$/K$_M$) of the mutant is 100-fold lower compared to the wild-type enzyme. On the other hand, the mutation has a rather similar and less pronounced effect on the catalytic efficiency of hydrolysis of the phosphoinositides, with a 8-, 7-, 5-, and 10-fold reduction for diC8-PI(4,5)P$_2$, diC8-PI(3,4,5)P$_3$, diC8-PI(3,5)P$_2$, and diC8-PI(5)P, respectively (**Figure 4—figure supplement 2**, **Figure 4—figure supplement 2—source data 1**). This decreased overall catalytic efficiency is due to an effect on both substrate binding (higher K$_M$) and turnover rate (smaller $k_{cat}$) for IP$_3$ and diC8-PI(4,5)P$_2$, while for diC8-PI(3,4,5)P$_3$ and diC8-PI(3,5)P$_2$ it can nearly completely be attributed to an effect on catalytic turnover. The observation that hydrolysis of all the phosphoinositides is affected to a similar degree suggests that the Y793C mutation indeed leads to an increased flexibility or conformational change of the entire P4IM-containing loop, potentially by disrupting the interactions with Y786 and P782.

Similar to Y793, the R800 residue is located in the P4IM, just before the conserved sequence motif P(A/S)W(C/T)DR(I/V)L. However, in contrast to the Y793 residue, its side chain is pointing into

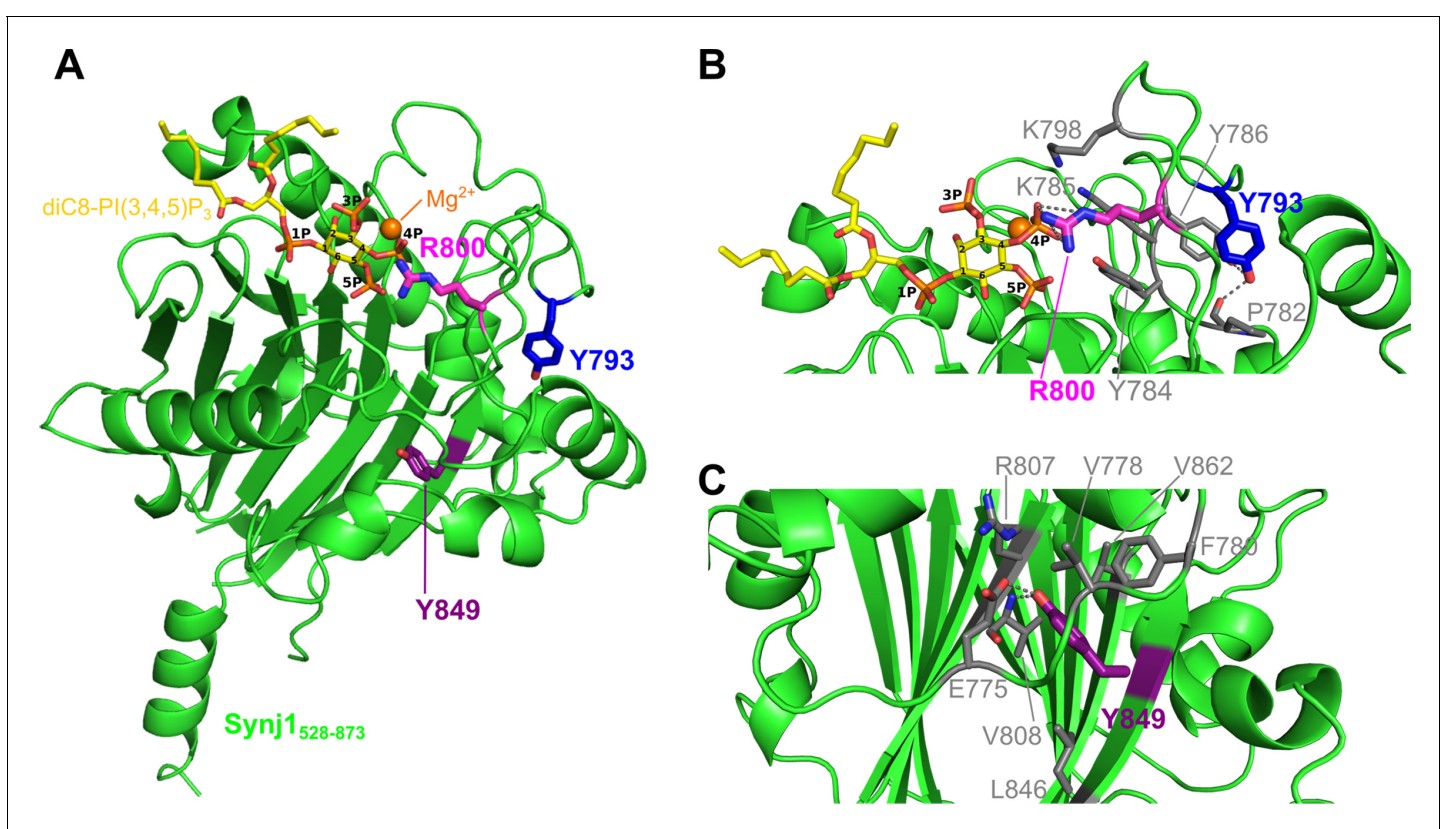

**Figure 5.** Localization of the Y793C, R800C, and Y849C disease mutations in the Synj1$_{528–873}$ structure. (**A**) Overall structure of Synj1$_{528–873}$ (green) with the Y793, R800, and Y849 residues represented as blue, magenta, and purple sticks, respectively. The Y793 residue is present in a loop close to the active site, while R800 is present in the active site. The Y849 residue, on the other hand, is buried in the core of the 5PPase domain. The Mg$^{2+}$-ion is represented as an orange sphere and the substrate, diC8-PI(3,4,5)P$_3$, as yellow sticks. (**B**) Close-up view on Y793 and R800 and their surrounding residues. Y793 forms a hydrogen bond with Y786 and with the main chain of P782 (grey dashes) to potentially stabilize the conformation of the loop. R800 forms multiple hydrogen bonds with the 4 P group of diC8-PI(3,4,5)P$_3$ (grey dashes). (**C**) Close-up view on Y849 and its surrounding residues. Y849 is buried in the hydrophobic core, where it forms a hydrogen bond with E775 and with the main chain NH of V808 (grey dashes).

the active site and directly interacts with the substrate via multiple (charged) hydrogen bonds with the 4 P group (*Figure 5A,B*). The clinical R800C mutation has a clear impact on the overall 5PPase activity ($k_{cat}/K_M$) of Synj1$_{528-873}$, with the extent of the effect varying depending on the substrate. The largest overall effect is also here observed for IP$_3$ with a 900-fold decrease of $k_{cat}/K_M$ due to the mutation, while the $k_{cat}/K_M$ value decreases 54-, 12-, 2-, and 1.2-fold for diC8-PI(4,5)P$_2$, diC8-PI(3,4,5)P$_3$, diC8-PI(3,5)P$_2$, and diC8-PI(5)P, respectively (*Figure 4—figure supplement 3A*, *Figure 4—figure supplement 3—source data 1*). From these values it is clear that the R800C mutation has a pronounced effect on the 5PPase reaction for substrates containing a phosphate group on position 4, while the mutation has almost no effect on the reaction for substrates without the 4 P group (*Figure 4—figure supplement 3B*). This effect on $k_{cat}/K_M$ is due to a combination of effects on catalytic turnover ($k_{cat}$) and binding ($K_M$). The mutation has a rather small but consistent effect on binding of the 4-P-containing substrates. On the other hand, the R800C mutation has a very high impact on the $k_{cat}$ value of IP$_3$, which could be caused by misalignment of this smaller substrate in the active site pocket if the interaction with the R800 sidechain is lost. Also for diC8-PI(4,5)P$_2$ a significant effect of the R800C mutation on $k_{cat}$ is observed, which indicates that R800 contributes to substrate turnover via its interaction with the 4 P group, potentially by properly aligning it for catalysis (see Discussion). Rather unexpectedly, only a small effect on $k_{cat}$ is observed for the R800C mutation using diC8-PI(3,4,5)P$_3$ as a substrate (*Figure 4—figure supplement 3A*).

The Y849 residue is buried for the largest part in the hydrophobic core of the protein on one of the β-strands (β12) forming the central β-sandwich fold of the 5PPase domain (*Figure 5A*, *Figure 1—figure supplement 1*). Here, its side chain is surrounded by residues G776, V778, F780, R807, L846 and V862, while its phenol hydroxyl groups forms H-bonds with E775 and the main chain amino-group of V808 (*Figure 5C*). In agreement with this location in the hydrophobic core, the Y849C mutant could not be expressed in the soluble fraction as a His-tagged protein, but a small amount of soluble protein could be obtained when expressing it as a GST-fusion. However, also here it was observed via size-exclusion chromatography that this protein eluted as a higher-oligomer, indicating a significant effect of the mutation on the protein fold and stability. Moreover, while it was confirmed that the wild-type Synj1$_{528-873}$ was fully active as a GST-fusion (data not shown), no activity could be found for the Y849C mutant at the highest enzyme concentration tested (1 μM) for any of the tested substrates (IP$_3$, diC8-PI(4,5)P$_2$, and diC8-PI(3,4,5)P$_3$) (*Figure 4—figure supplement 4*).

## Discussion

In this paper, we report the first structural information on the 5PPase domain of Synj1 (Synj1$_{528-873}$) to 2.3 Å resolution, relying on a strategy of Nanobody-aided crystallization to obtain well diffracting crystals. In addition to the Synj1$_{528-873}$ structure in the apo state, a short soak with diC8-PI(3,4,5)P$_3$ followed by flash freezing, also enabled us to trap this substrate in one of the active sites of the protein molecules present in the asymmetric unit, representing the very first structure of any 5PPase in complex with a genuine substrate. This structure thus reveals the interactions with all the important phosphate groups, including the scissile 5-phosphate group. Together with a detailed kinetic analysis of the contribution of these phosphate groups to catalysis, this allows us to propose a refined model for the catalytic mechanism of the 5PPase reaction extending on the previous models based on analogies with the apurinic/apyrimidinic base excision repair endonucleases (*Whisstock et al., 2000*; *Aboelnga and Wetmore, 2019*; *Dlakić, 2000*; *Mol et al., 2000*; *Erzberger and Wilson, 1999*), as shown in *Figure 6*.

In agreement with its conserved role in AP endonucleases and as previously suggested (*Trésaugues et al., 2014*), D730 is appropriately positioned to act as a general base by abstracting a proton from an attacking water molecule. Indeed, careful analysis of the electron density in the Synj1$_{528-873}$ diC8-PI(3,4,5)P$_3$-bound structure reveals electron density accounting for such a water molecule located at 3.1 Å from D730 and at 3.5 Å from the P$_5$-atom. This water molecule is furthermore held in place via a hydrogen bond with the conserved N732 residue (*Figure 3*; *Figure 6*). It should be noted that also the catalytically important H859 residue is located close to the nucleophilic water molecule, and, depending on the orientation of the side chain, H859 could either form an interaction with the O$_{PF}$-atom of the 5 P group or with this water molecule. The 5 P group is closely surrounded by multiple residues, H689, N732, R734, Y784, and H859, many of them potentially bearing a positive charge. Interactions of these residues with the 5 P group in the ground state

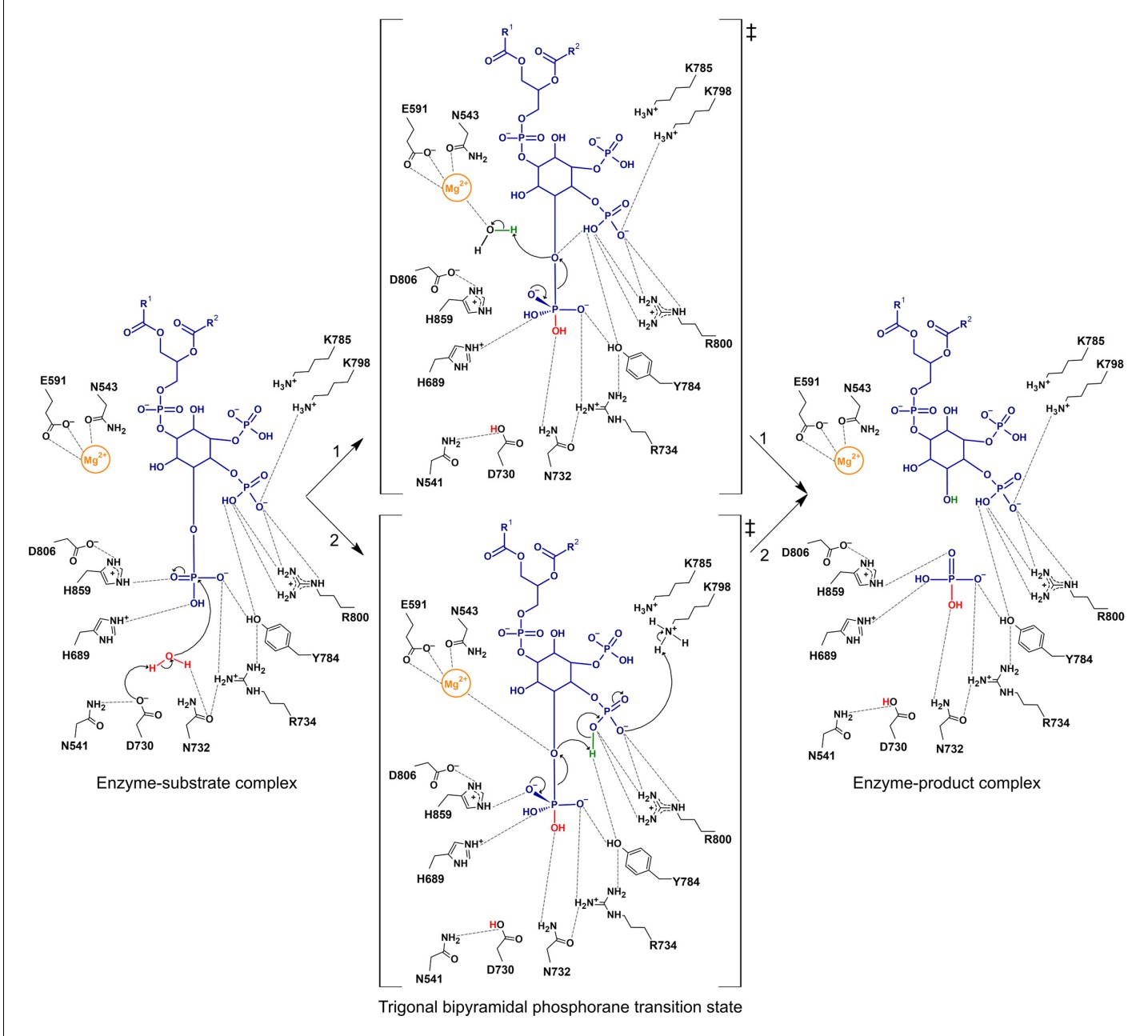

**Figure 6.** Proposed catalytic mechanism(s) of the 5-phosphatase reaction of Synj1. The nucleophilic water is activated by proton transfer to the catalytic base D730, allowing attack on the scissile phosphate ($P_5$), and resulting in a phosphorane transition state with excess negative charge that is stabilized by several surrounding residues. Two routes for leaving group activation are envisioned. In route 1 (upper pathway) a $Mg^{2+}$-activated water molecule acts as general acid by donating a proton to the leaving hydroxylate. Route 2 (lower pathway) corresponds to a mechanism of substrate-assisted catalysis, where the adjacent 4 P group acts as general acid, potentially assisted by transfer of a proton from K798.

contribute to binding, while stabilization of the excess of negative charge building up in the trigonal bipyramidal phosphorane transition state would contribute to catalysis. Previous mutagenesis studies on mouse INPP5A, yeast Inp52P and human INPP5B have shown that mutation of the residues corresponding to H689 and H859 negatively impact the activity, although from these studies it is not clear whether this is due to an effect on binding or catalysis (*Whisstock et al., 2000*; *Jefferson and Majerus, 1996*). While multiple interactions are formed with the 5 P group, none of the residues is seen within interaction distance of the oxygen bridging the scissile phosphate to the inositol ring ($O_5$) and

thus in an appropriate position to act as general acid to activate the leaving group. The $O_5$-atom is however located at a distance of 4.0 Å of the $Mg^{2+}$-ion allowing this distance to be bridged by a $Mg^{2+}$-coordinated water molecule as previously suggested (*Trésaugues et al., 2014*). Additionally, the $O_5$-atom is located in our structure at 3.9 Å from one of the oxygens ($O_{8P}$) of the 4 P group. This distance could decrease during the reaction path, allowing a direct role of the 4 P group in leaving group activation. We therefore envision two potential scenarios for leaving group activation to occur, indicated as route 1 and route 2 in *Figure 6*. Route 1 envisions a direct role of the $Mg^{2+}$-ion in leaving group activation by activating a $Mg^{2+}$-bound water molecule to donate a proton to the leaving $O_5$-atom, although such a water is not observed in our structure. A direct role of $Mg^{2+}$ in catalysis also agrees with its observed large contribution to catalytic turnover ($\Delta\Delta G_{catalysis}$ = 4.3 kcal/mol, *Figure 4*). In this scenario, the observed role of the 4 P group in catalysis could for example be due to a stabilizing hydrogen bond to the $O_5$-atom in the transition state (*Figure 6*). Route 2, on the other hand, corresponds to a mechanism of substrate-assisted catalysis, with a more direct role of the adjacent 4 P group in leaving group activation, where the $O_{8P}$ hydroxyl group of the 4 P group would transfer a proton to the leaving $O_5$-atom. The transferred proton could potentially originally come from the K798 residue in a proton relay mechanism (*Figure 6*). In our structure, the $O_{8P}$-atom from 4 P is located at 3.9 Å from the $O_5$-atom, but small rearrangements in going towards the transition state could easily bring both atoms in close contact. The residues surrounding the 4 P group could in turn contribute to catalysis by properly orienting the 4-phosphate for this role. Such a scenario is in good agreement with our kinetic data showing that the 4 P group has a large contribution to substrate turnover, with removal of the 4 P moiety (i.e. comparing diC8-PI(3,4,5)$P_3$ with diC8-PI(3,5)$P_2$) leading to a 340-fold reduction in $k_{cat}$ ($\Delta\Delta G_{catalysis}$ = 3.5 kcal/mol, *Figure 4*). Moreover, we also showed that the R800 residue contributes to catalysis via its interaction with the 4 P group. Within this scenario, the $Mg^{2+}$-ion could play its observed role in catalysis via a direct interaction with the negative charges building upon the $O_5$-atom in the transition state. It should of course be noted that the latter mechanism would not apply for the 5PPases that also efficiently catalyse the hydrolysis of PIPs missing the 4 P group (such as SHIP1, SHIP2, and to a lesser extent OCRL). At this point no conclusive distinction between these two scenarios can be made.

Apart from being a potential drug target for Alzheimer's disease, Down syndrome, and TBC1D24-linked epilepsy, loss-of-function mutations in Synj1 also lead to disease, including epilepsy and Parkinson's disease (*Hardies et al., 2016*; *Xie et al., 2019*; *Taghavi et al., 2018*; *Hong et al., 2019*; *Bouhouche et al., 2017*). This is an important point to take into account in a potential drug design effort. To rationalize such an effort and to find the available therapeutic window for inhibitory drug design, it is of paramount importance to characterize and quantify the effect of the disease-associated mutations on Synj1 activity in detail. At the moment of writing this manuscript, three disease-associated homozygous missense mutations had been described in the 5PPase domain of Synj1: Y793C, R800C, and Y849C (numbering based on Synaptojanin1-145 isoform 2) (*Hardies et al., 2016*; *Xie et al., 2019*; *Taghavi et al., 2018*). Our structural data and detailed kinetic analysis of the mutants in comparison to the wild-type enzyme allow us to gain deeper understanding in the molecular basis underlying the associated diseases. Our data shows that the Y849C mutant does not display any 5PPase activity with any of the tested substrates. The crystal structure shows that Y849 is nearly completely buried in the core of the protein (*Figure 5C*) and the Y849C mutation leads to severe problems in expressing this protein in a soluble form. Together, this indicates that the Y849C mutant corresponds to a near complete loss-of-function mutation due to disruption of the protein fold. In contrast, the R800 residue is located in the active site where it forms important interactions with the 4-phosphate group of the substrate (*Figure 5B*). Our kinetic analysis of the R800C mutant shows that this residue contributes to both substrate binding and catalysis, specifically via its interaction with the 4 P group. The contribution to substrate turnover ($k_{cat}$) can be explained by a model where R800 acts by orienting the 4 P group in an orientation suitable for subsequent proton transfer to the inositol 5-hydroxylate leaving group, as outlined above (*Figure 6*, route 2). Interestingly, the magnitude of the overall effect of the R800C mutation depends on the substrate that is being considered, with a decrease in $k_{cat}/K_M$ of a factor 900, 54, and 12 for $IP_3$, diC8-PI(4,5)$P_2$ and diC8-PI(3,4,5)$P_3$, respectively. Whether this difference in activity toward the different substrates is physiologically relevant and leads to an imbalance in the cellular distribution of PI(3,4,5)$P_3$, PI(4,5)$P_2$, and $IP_3$, remains to be seen. Such an imbalance of the PI(3,4,5)$P_3$/PI(4,5)$P_2$ ratio, with a relative increase in PI(4,5)$P_2$ versus PI(3,4,5)$P_3$ as expected from the kinetics of the R800C

mutant, has previously been linked to the occurrence of Parkinson's disease (*Sekar and Taghibiglou, 2018*). Finally, the Y793 residue is located on the same active site loop as R800, and likely contributes to stabilizing the loop conformation rather than being directly involved in interactions with the substrate (*Figure 5B*). Our kinetic analysis of the Y793C clinical mutant shows that the 5PPase activity of Synj1 is diminished between 5- and 10-fold for all the substrates in a rather indiscriminatory way.

As a general trend, it can be observed that the impact of the mutations on the catalytic efficiency of the Synj1$_{528-873}$ enzyme *in vitro* links with the severity and age of onset of the clinical manifestations in the patients that are homozygous carriers of these Synj1 mutations. Indeed, the very early onset of the severe neurodegeneration observed for the patients carrying a Y849C mutation (*Hardies et al., 2016*), corresponds well with our observation that this mutation leads to a complete loss of 5PPase activity. On the other hand, the R800C mutation leads to a decrease in overall activity of 54- to 12-fold for the most relevant substrates PI(4,5)P$_2$ and PI(3,4,5)P$_3$ and was reported to associate with parkinsonism and seizures at age 24 (*Taghavi et al., 2018*), while the even smaller effects on activity we observe for the Y793C mutation, with a decrease in activity of 8-fold for PI(4,5)P$_2$ and 7-fold for PI(3,4,5)P$_3$, leads to Parkinson's disease at later age (*Xie et al., 2019*). Although care should be taken with extending these observations to other mutations, this seems to indicate that the impact of the missense mutations on the kinetics of the 5PPase reaction *in vitro* can be used to a certain level to predict the severity of the disease outcome in patients who are homozygous for the corresponding mutation. Finally, these observations also have implications for the window that is available to therapeutically target the 5PPase domain of Synj1 in DS, AD, and TBC1D24-associated DOORS syndrome. While it was previously shown that genetic ablation of one Synj allele, corresponding to half of the normal cellular activity, was sufficient to rescue the disease phenotype in TBC1D24-mutant flies (*Fischer et al., 2016*), we show here that a close to 10-fold reduction of Synj1 activity on the long term could lead to disease in its own respect, thus still leaving a window for inhibitor design.

# Materials and methods

**Key resources table**

| Reagent type (species) or resource | Designation | Source or reference | Identifiers | Additional information |
|---|---|---|---|---|
| Gene (*Homo sapiens*) | SYNJ1 | NCBI | Gene ID: 8867 | |
| Strain, strain background (*Escherichia coli*) | BL21(DE3) pLysS | *Weiner et al., 1994* | Genotype: F⁻*hsdS*$_B$ (r$_B^-$m$_B^-$) *gal dcm* (DE3) pLysS (Cm$^R$) | Chemically (CaCl$_2$) competent |
| Strain, strain background (*E. coli*) | WK6 (Su⁻) | *Zell and Fritz, 1987* (PMID:3038536) | Genotype: Δ(*lac-proAB*) *galE strA*/F′ [lacI$^q$ lacZΔM15 proA$^+$B$^+$] | Chemically (CaCl$_2$) competent |
| Strain, strain background (M13 helper phage) | Kanamycin-resistant VCSM13 | Stratagene | 200251 | |
| Recombinant DNA reagent | pET28a (plasmid) | Novagen | 69864 | |
| Recombinant DNA reagent | pGEX-4T1 (plasmid) | GE Healthcare | GE28-9545-49 | |
| Recombinant DNA reagent | pMESy4 (plasmid) | *Pardon et al., 2014* (DOI: 10.1038/nprot.2014.039) | GenBank KF415192 | |
| Chemical compound, drug | IP$_6$ (D-*myo*-inositol 1,2,3,4,5,6-hexakis phosphate) | Merck Millipore | 407125 | |
| Chemical compound, drug | IP$_3$ (D-*myo*-inositol 1,4,5-trisphosphate) | Merck Millipore | 407137 | |

*Continued on next page*

*Continued*

| Reagent type (species) or resource | Designation | Source or reference | Identifiers | Additional information |
|---|---|---|---|---|
| Chemical compound, drug | diC8-PI(5)P | Echelon Biosciences | P-5008 | |
| Chemical compound, drug | diC8-PI(4,5)P$_2$ | Echelon Biosciences | P-4508 | |
| Chemical compound, drug | diC8-PI(3,5)P$_2$ | Echelon Biosciences | P-3508 | |
| Chemical compound, drug | diC8-PI(3,4,5)P$_3$ | Echelon Biosciences | P-3908 | |
| Chemical compound, drug | disodium-4-nitrophenyl phosphate (DNPP) | Sigma | N-4645 | |
| Sequence-based reagent | Y793C_F | This paper | PCR primers | CGACTGTGACACCA GTGAAAAGTGCCG |
| Sequenced-based reagent | Y793C_R | This paper | PCR primers | CTGGTGTCACAG TCGTCAGAAAACAAG |
| Sequence-based reagent | R800C_F | This paper | PCR primers | GTGCTGCACCCCTG CCTGGACAGAC |
| Sequenced-based reagent | R800C_R | This paper | PCR primers | GGTGCAGCACTTTT CACTGGTGTC |
| Sequence-based reagent | Y849C_F | This paper | PCR primers | CACTGTGGAAGAG CTGAGCTGAAG |
| Sequenced-based reagent | Y849C_R | This paper | PCR primers | CTTCCACAGTGCAGC AAAGTGCCTGG |
| Peptide, recombinant protein | CaptureSelect Biotin anti-C-tag conjugate | Thermo Fisher Scientific | 7103252100 | |
| Peptide, recombinant protein | Streptavidin Alkaline Phosphatase | Promega | V5591 | |
| Commercial assay or kit | Malachite Green Phosphate Assay kit | Gentaur | POMG-25H | |
| Software, algorithm | autoPROC | *Vonrhein et al., 2011* (DOI: 10.1107/S0907444911007773) | RRID:SCR_015748 https://www.global phasing.comautoproc/ | |
| Software, algorithm | STARANISO | *Tickle et al., 2018* | RRID:SCR_018362 http://staraniso.globalphasing .org/cgi-bin/staraniso.cgi | |
| Software, algorithm | Phaser | *McCoy et al., 2007* (DOI:10.1107/S0021889807021206) | RRID:SCR_014219 https://www.phenix-online.org/ documentation/reference/phaser.html | |
| Software, algorithm | Phenix.Ligand Fit | *Terwilliger et al., 2006* (DOI:10.1107/S0907444906017161) | https://www.phenix-online.org/ documentation/reference/ligandfit.html | |
| Software, algorithm | Phenix.Refine | *Afonine et al., 2012* (DOI: 10.1107/S0907444912001308) | RRID:SCR_016736 https://www.phenix-online.org/ documentation/reference/ refinement.html | |
| Software, algorithm | Coot | *Emsley et al., 2010* (DOI: 10.1107/S0907444910007493) | RRID:SCR_014222 https://www2.mrc-lmb.cam.ac.uk/ personal/pemsley/coot/ | |
| Software, algorithm | MolProbity | *Chen et al., 2010* (DOI: 10.1107/S0907444909042073) | RRID:SCR_014226 http://molprobity. biochem.duke.edu | |
| Software, algorithm | PDB-REDO server | *Joosten et al., 2014* (DOI: 10.1107/S2052252514009324) | RRID:SCR_018936 https://pdb-redo.eu/ | |

*Continued on next page*

*Continued*

| Reagent type (species) or resource | Designation | Source or reference | Identifiers | Additional information |
|---|---|---|---|---|
| Software, algorithm | PyMOL (version 2.0) | Schrödinger | RRID:SCR_000305 https://pymol.org/2/ | |
| Software, algorithm | GraphPad Prism (version 8) | Graphpad Software | RRID:SCR_002798 | |
| Software, algorithm | CCP4 suite | *Winn et al., 2011* (DOI: 10.1107/S0907444910045749) | RRID:SCR_007255 http://www.ccp4.ac.uk/ | |
| Software, algorithm | Clustal Omega | *Madeira et al., 2019* (DOI: 10.1093/nar/gkz268) | RRID:SCR_001591 http://www.ebi.ac.uk/ Tools/msa/clustalo/ | |
| Software, algorithm | ESPript | *Robert and Gouet, 2014* (DOI: 10.1093/nar/gku316) | RRID:SCR_006587 http://espript.ibcp.fr /ESPript/ESPript/ | |
| Software, algorithm | ACD/ChemSketch (version 2019.2.1) | Advanced Chemistry Development | http://www.acdlabs.com | |

## Cloning, protein expression, and protein purification

The open reading frame (ORF) encoding the 5-phosphatase domain of Synj1 (residues 528–873) was amplified from the full length Synj1 ORF (NCBI - Gene ID: 8867) and an N-terminal TEV cleavage site was added. This PCR product was digested with NdeI and NotI and ligated into a pET28a expression vector (Novagen). The Synj1$_{528-873}$ ORF was also amplified and inserted into a pGEX-4T1 expression vector (GE Healthcare) containing a pre-inserted TEV-site using the BamHI and NotI sites (GE Healthcare). QuickChange site-directed mutagenesis was used to insert the Y793C and R800C mutations into the pET28a-TEV-Synj1$_{528-873}$ plasmid, while the Y849C mutation was introduced in the pGEX-4T1-nTEV-Synj1$_{528-873}$ plasmid. The resulting plasmids were verified by sequencing (Euro-fins Genomics).

Plasmids containing the wild-type or mutant ORFs were transformed in *E. coli* BL21 (DE3) pLysS cells. Cells were grown at 37°C in Terrific Broth (TB) medium supplemented with the appropriate antibiotics until an $OD_{600}$ of 0.6 was reached. Subsequently, protein expression was induced by addition of 1 mM IPTG and after an incubation period of 17 hr at 20°C, cells were harvested by centrifugation.

All steps of Synj1$_{528-873}$ purification (WT and mutants) were performed at 4°C. The bacterial pellets containing His-tagged wild-type, Y793C, and R800C proteins were resuspended in buffer A (25 mM HEPES pH 7.5, 300 mM NaCl, 5% glycerol, and 5 mM $MgCl_2$) supplemented with 10 mM imidazole pH 8, 1 mM DTT, 1 µg/ml leupeptin protease inhibitor (Roth), 0.1 mg/ml AEBSF serine protease inhibitor (Roth), 2 µM pepstatin A (Promega) and 50 µg/ml DNaseI (Sigma), and lysed with a cell-disruptor system (Constant Systems). After clearance of the lysate via centrifugation, the supernatant was loaded onto a $Ni^{2+}$-NTA-Sepharose column (GE Healthcare) equilibrated with buffer A supplemented with 10 mM imidazole pH 8. After extensive washing, the protein was eluted by increasing the imidazole concentration to 1 M and fractions containing the protein of interest were pooled. To cleave the His-tag, 1 mg of His-tagged TEV protease was added per 10 mg of His-Synj1$_{528-873}$ and the mixture was dialysed overnight against buffer A supplemented with 1 mM DTT. The mixture was then loaded onto a $Ni^{2+}$-NTA-Sepharose column, to remove the His-tagged TEV protease and any remaining non-cleaved protein. The bacterial pellets containing GST-tagged wild-type and Y849C protein were resuspended in buffer B (25 mM HEPES pH 7.5, 150 mM NaCl, 5% glycerol and 5 mM $MgCl_2$) supplemented with 1 mM DTT, 1 µg/ml leupeptin protease inhibitor, 0.1 mg/ml AEBSF serine protease inhibitor, 2 µM pepstatin A and 50 µg/ml DNaseI, and cells were lysed as before. The cell lysate was cleared by centrifugation and the supernatant was incubated with Glutathione Sepharose 4 Fast Flow beads (GE Healthcare) for 1 hr, then packed into an empty PD-10 column (GE Healthcare). Following extensive washing, the protein was eluted with buffer B supplemented with 10 mM of reduced glutathione. As a final purification step, the His-tagged or tag-less proteins were loaded on a Superdex 75 s column (GE Healthcare), while the GST-tagged proteins

were loaded on a Superdex 200 s column (GE Healthcare), using buffer B supplemented with 1 mM DTT as running buffer.

## Nanobody (Nb) generation and purification

A llama was immunized with His-Synj1$_{528-873}$. A six-week immunization protocol was followed consisting of weekly immunizations of 200 µg (first two weeks) or 100 µg (last four weeks) protein in presence of GERBU adjuvant. All animal vaccinations were performed in strict accordance with good practices and EU animal welfare legislation. Blood was collected four days after the last injection. Library construction, Nb selection via phage display and Nb expression and purification were performed as described previously (*Pardon et al., 2014*). Briefly, the variable domains of the heavy-chain antibody repertoire from the llama were subcloned in a pMESy4 phage display vector, which adds a C-terminal His-tag and EPEA-tag (=CaptureSelect C-tag). This resulted in an immune library of 2.4•10$^9$ transformants. This Nb-repertoire was expressed on phages after rescue with the VCSM13 helper phage, and two consecutive rounds of phage display were used to select for phages expressing Nbs that bind to the 5PPase domain of Synj1. Therefore, two different coating strategies were used. In the first coating strategy, biotinylated Synj1$_{528-873}$ (as well His-tagged as non-tagged) was captured on neutravidine-coated 96 well-plates and all binding and washing steps were performed in buffer B (25 mM HEPES pH 7.5, 150 mM NaCl, 5% glycerol and 5 mM MgCl$_2$). In the second coating strategy, Synj1$_{528-873}$ was captured directly on the bottom of a 96-well plate and all binding and washing steps were performed in buffer B supplemented with 1 mM DTT. After phage display selection, an ELISA screen was performed on crude cell lysates of *E. coli* expressing the Nbs, in order to confirm binding. Synj1$_{528-873}$ was coated on the ELISA plate. Incubation with the Nb-containing cell extracts and all washing steps were performed in buffer B. Binding of the Nbs was detected via their EPEA-tag using a 1:4000 CaptureSelect Biotin anti-C-tag conjugate (Thermo Fisher Scientific) in combination with 1:1000 Streptavidin Alkaline Phosphatase (Promega). Colour was developed by adding 100 µl of a 3 mg/ml disodium-4-nitrophenyl phosphate solution (DNPP, Sigma) and measured at 405 nm. Sequence analysis was used to classify the binding Nb clones in sequence families.

For Nb production and purification, pMESy4 vectors, containing the Nb ORFs, were transformed in *E. coli* WK6 (Su⁻) cells. Cells were grown at 37°C in TB medium supplemented with 100 µg/ml ampicillin, 0.1% glucose, and 2 mM MgCl$_2$, until an OD$_{600}$ of 0.6 was reached. Nb expression was induced by adding 1 mM IPTG. After incubation for 17 hr at 28°C, cells were harvested by centrifugation and subjected to an osmotic shock to obtain the periplasmic extract. Subsequently, an affinity purification step on Ni$^{2+}$-NTA sepharose and a SEC step on a Superdex 75 16/60 column (in buffer C: 25 mM HEPES pH 7.5, 150 mM NaCl, 5% glycerol) were used to purify the Nbs.

## Crystallization and data collection

To form the Nb-Synj1$_{528-873}$ complex, 250 µM of Synj1$_{528-873}$ was mixed with 500 µM of Nb and incubated for 1 hr on ice. Subsequently, a Superdex 75 10/30 column (in buffer B: 25 mM HEPES pH 7.5, 150 mM NaCl, 5% glycerol, and 5 mM MgCl$_2$) was used to separate the complex from the excess of Nb.

Initial crystallization conditions were found using the Wizard III and IV (Rigaku) and SG1 screen (Molecular Dimensions) by the sitting-drop vapour-diffusion method at 20°C using a Mosquito robot (SPT Labtech). After optimization two similar conditions yielded good quality crystals of the Nb15-Synj1$_{528-873}$ complex prepared at 25 mg/ml. Condition 1 was composed of 15% PEG 4000, 0.1 M sodium citrate pH 5 and 10% 2-propanol, while condition 2 contained 15% PEG 3350, 0.1 M sodium citrate pH 5.5 and 13% ethanol. Crystals from these conditions were soaked overnight with mother liquor supplemented with either 1 mM IP$_6$ (inositol-(1,2,3,4,5,6)-hexakisphosphate) (Merck Millipore) (condition 1) or 1 mM diC8-PI(3,4,5)P$_3$ (Echelon Biosciences) (condition 2). Subsequently, the former crystals were transferred to a cryo-solution containing mother liquor supplemented with 25% glycerol, while the latter crystals were again very briefly soaked in mother liquid supplemented with 1 mM diC8-PI(3,4,5)P$_3$ and 15% glycerol as cryo-protectant, immediately followed by flash freezing in liquid nitrogen.

All data were collected at 100 K. Diffraction data from the crystal soaked with IP$_6$ was collected at the i03 beamline of the DIAMOND synchrotron (λ = 0.980105) using an Eiger2 XE 16M detector.

Data from the crystal soaked with diC8-PI(3,4,5)$P_3$ was collected at the Proxima 2a beamline of the SOLEIL synchrotron ($\lambda$ = 0.976246) equipped with an Eiger X 9M detector. Diffraction data were integrated and scaled with autoPROC (Global Phasing Limited; *Vonrhein et al., 2011*), using the default pipeline which includes XDS, Truncate, Aimless, and STARANISO (*Tickle et al., 2018*). Anisotropy analysis by STARANISO showed that diffraction data were anisotropic, with diffraction limits along the reciprocal axes of 2.71 Å along 0.781 a* - 0.625 c*, 2.41 Å along b* and 2.30 Å along 0.975 a* + 0.222 c* for the structure obtained from the $IP_6$-soaked crystal (*Appendix 1—figure 1A*), and 2.86 Å along 0.043 a* + 0.999 c*, 2.73 Å along b* and 3.14 Å along −0.974 a* + 0.228 c* for the structure obtained from the diC8-PI(3,4,5)$P_3$-soaked crystal (*Appendix 1—figure 1B*). Automated resolution cutoff of anisotropic corrected data by STARANISO resulted in a 2.30 Å and a 2.73 Å resolution structure for the $IP_6$- and diC8-PI(3,4,5)$P_3$-soaked crystals respectively (using I/σ(I) > 1.4 as a cut-off criterion).

## Structure determination and refinement

For the $IP_6$ soaked crystal the phase problem was solved by molecular replacement using Phaser (*McCoy et al., 2007*) from the PHENIX suit (*Liebschner et al., 2019*). The structures of other 5-phosphatase domains (namely the 5-phosphatase domains of *Schizosaccaromyces pombe* Synaptojanin, human INPP5B, human INPP5B in complex with diC8-PI(4)P and human OCRL) and a random Nb were used as search models (PDB entries 5-phosphatases: 1i9y, 3n9v, 3mtc and 4cmn; PDB entry Nb: 4nc2). Since the resulting structure had no $IP_6$ bound, we flagged it as an apo-structure. The structure of the diC8-PI(3,4,5)$P_3$-containing crystal was solved by refining the data against the refined apo-structure, taking care to use the same set of reflections for cross-validation. Analysis of the 2Fo-Fc, Fo-Fc, and omit maps revealed unambiguous electron density for diC8-PI(3,4,5)$P_3$. Subsequently, the ligand was inserted using Phenix.LigandFit (*Terwilliger et al., 2006*) followed by a refinement using Phenix.Refine (*Afonine et al., 2012*).

Models were improved by iterative cycles of refinement with Phenix.Refine (*Afonine et al., 2012*) and manual building in Coot (*Emsley et al., 2010*). MolProbity (*Chen et al., 2010*) was used for structure validation. As a final optimization, the diC8-PI(3,4,5)$P_3$-bound structure was submitted to the PDB-REDO server (*Joosten et al., 2014*). X-ray data collection and refinement statistics are listed in *Table 1*.

Coordinates and structure factors have been deposited in the Protein Data Bank under accession codes PDB 7A0V (apo Synj1$_{528-873}$) and PDB 7A17 (diC8-PI(3,4,5)$P_3$-bound Synj1$_{528-873}$).

## Structural analysis

All structural figures were produced with PyMOL (The PyMol Molecular Graphic System, version 2.0 Schrödinger, LLC, https://pymol.org/2/). Superpose in CCP4 (*Krissinel and Henrick, 2004*) was used to determine the root-mean-square deviation (rmsd) between the different Synj$_{528-873}$ chains present in one asymmetric unit, and between one Synj1$_{528-873}$ chain and the structures of the 5PPase domain of the other 5PPases. Multiple sequence alignment of the 5PPase domains of all human 5PPases and SPSynj was performed via Clustal Omega (*Madeira et al., 2019*). ESPript was used to assign secondary structures (*Robert and Gouet, 2014*). ACD/ChemSketch (version 2019.2.1, Advanced Chemistry Development, Inc, Toronto, ON, Canada, http://www.acdlabs.com, 2020) was used to draw the schematic representations of the interactions in the active site.

## Enzyme kinetics

The enzymatic activity of wild-type and mutant Synj1$_{528-873}$ using the substrates $IP_3$ (Sigma), diC8-PI(3,4,5)$P_3$, diC8-PI(4,5)$P_2$, diC8-PI(3,5)$P_2$, and diC8-PI(5)P (Echelon Biosciences) was measured using the Malachite Green phosphate assay (Gentaur) that detects the release of free orthophosphates ($P_i$). A full steady-state Michaelis-Menten analysis of Synj1$_{528-873}$ wild-type and the R793C and R800C mutants was performed via initial rate measurements using an enzyme concentration optimized for each enzyme/substrate combination to convert around 10% of substrate over the total measuring time, and at varying substrate concentrations (typically within the range 5–500 μM, depending on the $K_M$ value). For the GST-tagged Y849C mutant, time measurements at a single substrate concentration of 120 μM (for $IP_3$, diC8-PI(4,5)$P_2$, and diC8-PI(3,4,5)$P_3$) and an enzyme concentration of 1 μM were performed. To assess the effect of Nb15 on catalysis, measurements were

carried out by incubating Synj1$_{528-873}$ with a 40-fold excess (100 nM) of Nb15 for 10 min at 25°C prior to the assay. All measurements were done at 25°C in 25 mM HEPES pH 7.5, 150 mM NaCl, 5% glycerol, 2 mM MgCl$_2$ and 1 mM DTT, except for the measurements performed at pH 5.5 where the HEPES was replaced by 25 mM sodium citrate pH 5.5. At different time points, 80 µl of each reaction mixture was transferred to a 96 well plate containing 20 µl of the Malachite Green working reagent to stop the reaction. After 30 min of incubation, the absorption of the samples was measured in a SPECTROstar$^{Nano}$ (BMG Labtech) plate reader at 620 nm. The absorption was plotted against time and a linear trendline was drawn through the plotted points. To obtain the initial reaction velocity (v), the slopes were divided by the slope of a standard curve (measured in quadruplicate). The velocity divided by the enzyme concentration (v/E$_0$) was plotted against the substrate concentration and the curve was fitted on the Michaelis-Menten equation in GraphPad Prism (version 8, GraphPad Software, La Jolla, California USA, http://www.graphpad.com) to determine k$_{cat}$ and K$_M$. Each datapoint was measured in triplicate. ΔΔG values were calculated as follows: $\Delta\Delta G_{overall}$ = -R.T. ln ([k$_{cat}$/K$_M$]$_{substrate\ 2}$/[k$_{cat}$/K$_M$]$_{substrate\ 1}$); $\Delta\Delta G_{binding}$ = -R.T. ln ([1/K$_M$]$_{substrate\ 2}$/[1/K$_M$]$_{substrate\ 1}$); $\Delta\Delta G_{calatysis}$ = -R.T. ln ([k$_{cat}$]$_{substrate\ 2}$/[k$_{cat}$]$_{substrate\ 1}$).

## Acknowledgements

We thank the staff at the beamlines Proxima 2a of the Soleil synchrotron (France) and i03 of the Diamond synchrotron (United Kingdom) for assistance during data collection. We thank the members of the Versées lab for comments and discussions. This work was supported by the Fonds voor Wetenschappelijk Onderzoek (FWO grant number G0D3317N to PV and WV) and Strategic Research Program Financing from the VUB (WV). JP, EM and BD received a fellowship from the Fonds voor Wetenschappelijk Onderzoek. JP and CG solved the structures. JP, EM, BD, MB, RS and YL produced and purified proteins and performed biochemical experiments. EP, JS, PV and WV supervised experiments and aided in interpreting data. WV designed the study. JP, EM, CG and WV wrote the manuscript. All authors reviewed the manuscript.

## Additional information

### Competing interests

Patrik Verstreken: Reviewing editor, *eLife*. The other authors declare that no competing interests exist.

### Funding

| Funder | Grant reference number | Author |
|---|---|---|
| Fonds Wetenschappelijk Onderzoek | 1S04918N | Jone Paesmans |
| Fonds Wetenschappelijk Onderzoek | 1S09120N | Ella Martin |
| Fonds Wetenschappelijk Onderzoek | G0D3317N | Patrik Verstreken Wim Versées |
| Fonds Wetenschappelijk Onderzoek | 11D4621N | Babette Deckers |
| Vrije Universiteit Brussel | Strategic Research Program Financing - SRP50 | Wim Versées |

The funders had no role in study design, data collection and interpretation, or the decision to submit the work for publication.

### Author contributions

Jone Paesmans, Ella Martin, Data curation, Formal analysis, Investigation, Writing - original draft, Writing - review and editing; Babette Deckers, Data curation, Formal analysis, Investigation, Writing - review and editing; Marjolijn Berghmans, Ritika Sethi, Yannick Loeys, Formal analysis, Investigation,

Writing - review and editing; Els Pardon, Jan Steyaert, Patrik Verstreken, Supervision, Methodology, Writing - review and editing; Christian Galicia, Formal analysis, Supervision, Investigation, Writing - original draft, Writing - review and editing; Wim Versées, Conceptualization, Formal analysis, Supervision, Funding acquisition, Investigation, Writing - original draft, Project administration, Writing - review and editing

## Author ORCIDs
Jone Paesmans  https://orcid.org/0000-0002-3292-4609
Ella Martin  https://orcid.org/0000-0002-9607-7074
Babette Deckers  https://orcid.org/0000-0002-3855-4776
Marjolijn Berghmans  http://orcid.org/0000-0002-8699-6915
Els Pardon  http://orcid.org/0000-0002-2466-0172
Jan Steyaert  http://orcid.org/0000-0002-3825-874X
Christian Galicia  https://orcid.org/0000-0001-6080-7533
Wim Versées  https://orcid.org/0000-0002-4695-696X

## Decision letter and Author response
Decision letter https://doi.org/10.7554/eLife.64922.sa1
Author response https://doi.org/10.7554/eLife.64922.sa2

## Additional files
### Supplementary files
• Supplementary file 1. (**A**) Comparison of the 5PPase domain of human Synj1 (Synj1$_{528–873}$) with the corresponding 5PPase domain of the other human inositol polyphosphate 5-phosphatases and SPSynj. The root-mean-square deviation (rmsd) after superposition of the structures was determined using CCP4 SUPERPOSE. Furthermore, the sequence identity was determined using Clustal Omega. (**B**) Residues interacting with the diC8-PI(3,4,5)P$_3$ substrate in Synj1$_{528–873}$ and the corresponding residues in the other nine human 5PPases and SPSynj. Completely conserved residues are written in white with a red background and similar residues are written in red.

• Transparent reporting form

### Data availability
Diffraction data have been deposited in the PDB under the accession code 7A0V and 7A17. All data generated or analysed during this study are included in the manuscript and supporting files. Source data files have been provided for Figure 4 and Figure 4—figure supplements 1–3.

The following datasets were generated:

| Author(s) | Year | Dataset title | Dataset URL | Database and Identifier |
|---|---|---|---|---|
| Paesmans J, Galicia C, Martin E, Versées W | 2020 | Crystal structure of the 5-phosphatase domain of Synaptojanin1 in complex with a nanobody | https://www.rcsb.org/structure/7A0V | RCSB Protein Data Bank, 7A0V |
| Paesmans J, Galicia C, Martin E, Versées W | 2020 | Crystal structure of the 5-phosphatase domain of Synaptojanin1 bound to its substrate diC8-PI(3,4,5)P$_3$ in complex with a nanobody | https://www.rcsb.org/structure/7A17 | RCSB Protein Data Bank, 7A17 |

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

## Appendix 1

### Anisotropy analysis

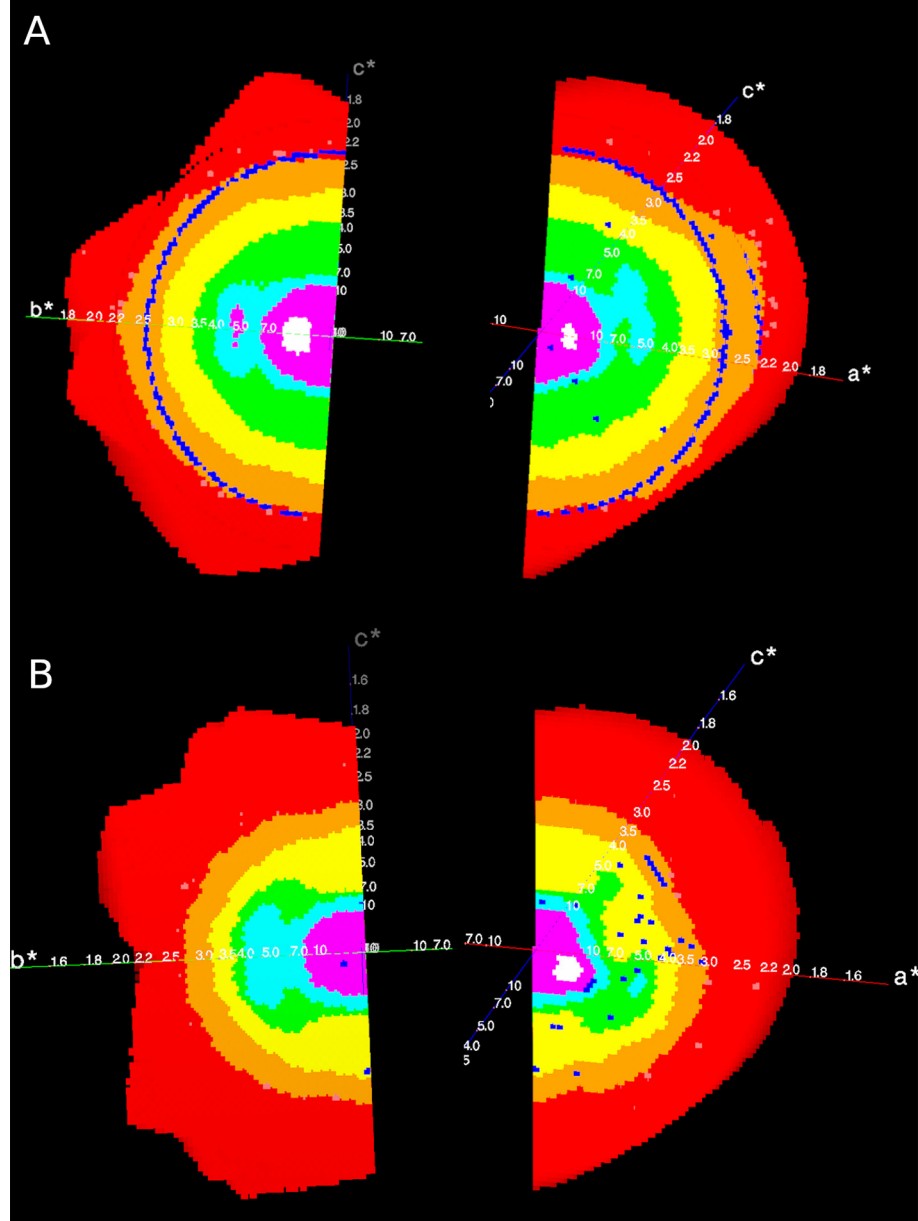

**Appendix 1—figure 1.** Reciprocal lattices (axes a*, b*, c*) colour coded by mean I/σ(I) as given by STARANISO, showing the anisotropic diffraction of the crystal of (A) the apo Synj1$_{528-873}$ and (B) the diC8-PI(3,4,5)P$_3$-bound Synj1$_{528-873}$.

