## [Decision Letter]

**Acceptance summary:**

Paesmans and colleagues describe the nanobody-bound crystal structure of the 5-phosphatase domain of human synaptojanin1, an important regulator of clathrin-mediated endocytosis, with and without a trapped substrate. Based on the structure, they kinetically characterize catalysis towards different substrates and come up with two alternative mechanisms how Synj1 mediates catalysis. They finally characterize three disease mutants of synaptojanin1, which show reduced activity or aggregate.

**Decision letter after peer review:**

[Editors’ note: the authors submitted for reconsideration following the decision after peer review. What follows is the decision letter after the first round of review. Please note that in a second round of review, the reviewers requested the chain order be changed, so that chain E became chain A.]

Thank you for submitting your work entitled "A structure of substrate-bound Synaptojanin1 provides new insights in its mechanism and the effect of disease mutations" for consideration by *eLife*. Your article has been reviewed by three peer reviewers, one of whom is a member of our Board of Reviewing Editors, and the evaluation has been overseen by a Senior Editor. The reviewers have opted to remain anonymous.

Our decision has been reached after consultation between the reviewers. Based on these discussions and the individual reviews below, we regret to inform you that your work will not be considered further for publication in *eLife*.

The reviewers are concerned that the crystallized substrate-bound complex is non-productive. Moreover, the absence of substrate in 2 of the 3 synaptojanin molecules is troublesome, and there are other concerns about this structure (details in reviews below). Thus, in the present form, the paper is not acceptable.

Reviewer #1:

This is a potentially a major contribution towards understanding the molecular mechanism of synaptojanin. The previous structure of the 5-phosphatase domain of *S. pombe* synaptojanin in complex with inositol‐(1,4)‐bisphosphate (Tsujishita, et al., PDB ID 1i9z) was determined at high resolution but the electron density for the inositol-(1,4)-bisphosphate was ambiguous. As the authors point out, subsequent structures of the catalytic 102 domain of INPP5B in complex with diC8‐PI(4)P and diC8‐PI(3,4)P2 revealed another orientation but these structures lack the 5-phosphate group. In this work the authors present high-resolution structures of the 5-phosphatase domain of synaptojanin, and its complex with the substrate diC8-PI(3,4,5)P3, providing a high-resolution image of a 5-phosphatase with a trapped substrate in its active site. The new structures suggest an enzymatic mechanism and explain how certain disease-associated mutations affect the catalytic rate.

In the substrate-bound structure, electron density for diC8-PI(3,4,5)P3 was only found for 1 of the three molecules in the asymmetric unit. Electron density consistent with phosphate groups was found for the other two molecules. However, the B-factors are perhaps a little higher around the binding site for one of the two molecules without substrate. Is it possible that substrate was partially occupied in that site? Please provide simulated annealing omit maps for all three binding sites.

A cis-peptide bond is found between Y784 and K785, as in all other 5-phosphatase structures except in the deposited structure of *S. pombe* synaptojanin. It might be instructive to try to fit a cis peptide bond against the *S. pombe* diffraction data.

As the authors point out, the effect of the Y793C mutant might be indirect by affecting nearby residues at the active site. It is particularly interesting that Y793 is close to the cis-peptide bond between Y784 and K785. Is it possible that the mutation destabilizes this cis-peptide bond, corroborating its functional importance?

Reviewer #2:

Paesmans and colleagues describe the nanobody-bound crystal structure of the 5-phosphatase domain of human Synj1, an important regulator of clathrin-mediated endocytosis, with and without a trapped substrate. Based on the structure, they kinetically characterize catalysis towards different substrates and come up with two alternative mechanisms how Syn1 mediates catalysis. They finally characterize three disease mutants of Syn1, which show reduced activity or aggregate.

The paper is well drafted, the nanobody-aided structure determination of Syn1 is elegant and the kinetic experiments appear overall sound. A question to discuss is whether the novelty of the structural data is sufficient for publication in e*Life*. The structure of yeast synaptojanin is known for a long time and it seems rather similar to the human counterpart (although this is difficult to judge from this manuscript). Furthermore, many features of substrate binding and catalysis were already deduced from the product-bound yeast synaptojanin structure. I have also some questions/doubts about the substrate-bound structure.

1) Detailed structural comparisons of Syn1 to other 5'-phosphatases (including rmsd) are missing to judge similarities and recognize potential differences between the structures. A structural comparison of the catalytic centers of some 5' phosphatases, including human and yeast synaptojanin with bound ligands, complementing Supplementary file 1, would allow the reader to appreciate (potentially) new mechanistic insights from the Syn1 structure.

2) A major point of the paper is the substrate bound to Syn1. However, it remains unclear why the substrate has not been turned over after an overnight incubation at 20 C. The authors speculate about a missing catalytic water, but this cannot explain missing catalysis since the substrate-binding site is apparently accessible, in particular for water. Is the crystallization condition incompatible with catalysis? Does nanobody-binding reduce catalysis? Is there at all any substrate left after overnight incubation with the crystals? Without answering this, it remains uncertain whether the observed ligand is indeed a non-converted substrate molecule (see also next point).

3) Crystallographic data of the substrate-bound crystals: Why is the signal to noise value so low for the substrate-bound crystals? Why did the authors apply anisotropic corrections to their data? Did the crystals diffracted anisotropically – if yes, provide some more information, e.g. to which resolution along a*,b*,c*. Judging from the spherical completeness, the dataset rather correspond to a 3.x A resolution dataset in terms of reflection numbers. It is astonishing that for such low resolution, 91 water molecules with an average B-factor of only 34 Angstroms could be modelled (I would hardly expect to see any visible water molecules at this resolution). Is the model overfitted? Any explanation for the wide gap of Rwork and Rfree? What was the exact refinement strategy? Overall, the data quality of the substrate-bound crystals is borderline to accurately evaluate ligand binding, in particular if the identity of the bound ligand is not 100% clear.

Reviewer #3:

The authors report crystal structures of the apo and substrate bound versions of the inositol 5-phosphatase domain of human synaptojanin-1 (Synj1), crystallized in the presence of a nanobody. The overall structure is similar to those of several previously solved 5-phosphatase enyzmes from yeast and humans. The main addition to the field in the current structure with respect to past work is that a substrate is present in this case, whereas previous structures were apo or product-bound.

The structure contains 3 catalytic chains per asymmetric unit. The ligand-bound form was obtained by soaking apo crystals in a di-C8 form of the substrate phosphoinositide. The structure of the complex was determined at 2.9 Å resolution. Detail on the catalytic geometry is limited at this resolution. Water and metal ion positions are not visualized at all, or with limited accuracy. Only one of the three catalytic chains binds to substrate. I did not see an explanation for the lack of binding to the other two chains. The 5-phosphate is still present on the substrate, despite that catalytically active wild-type enzyme was used. The authors present a detailed scheme for the catalytic mechanism on the basis of their structure, but I did not see an explanation for the failure of the enzyme to hydrolyze the 5-phosphate. I would speculate that the pH 5.0 of the crystallization could account for this. Have the authors attempted to adapt the crystals to neutral pH? Is a non-hydrolyzable substrate analogue available, or could it be synthesized? (I appreciate that phosphoinositide synthetic chemistry is non-trivial.) It would certainly be helpful for the field to know the geometry of a catalytically productive substrate complex, yet I am not convinced that the authors have as yet succeeded in obtaining such a structure. In my opinion, the conclusions concerning enzyme mechanism are, at this stage, over-interpreted.

The paper is, apart from the omissions noted above, very thorough. It reads like a review article in the Introduction and Discussion. The results includes more detail than is standard. Overall the paper would be more readable if it were edited to about half of the current length.

---

## [Author Response]

[Editors’ note: the authors resubmitted a revised version of the paper for consideration. What follows is the authors’ response to the first round of review. Please note that in a second round of review, the reviewers requested the chain order be changed, so that chain E became chain A.]

Reviewer #1:This is a potentially a major contribution towards understanding the molecular mechanism of synaptojanin. The previous structure of the 5-phosphatase domain of *S. pombe* synaptojanin in complex with inositol‐(1,4)‐bisphosphate (Tsujishita, et al., PDB ID 1i9z) was determined at high resolution but the electron density for the inositol-(1,4)-bisphosphate was ambiguous. As the authors point out, subsequent structures of the catalytic 102 domain of INPP5B in complex with diC8‐PI(4)P and diC8‐PI(3,4)P2 revealed another orientation but these structures lack the 5-phosphate group. In this work the authors present high-resolution structures of the 5-phosphatase domain of synaptojanin, and its complex with the substrate diC8-PI(3,4,5)P3, providing a high-resolution image of a 5-phosphatase with a trapped substrate in its active site. The new structures suggest an enzymatic mechanism and explain how certain disease-associated mutations affect the catalytic rate.In the substrate-bound structure, electron density for diC8-PI(3,4,5)P3 was only found for 1 of the three molecules in the asymmetric unit. Electron density consistent with phosphate groups was found for the other two molecules. However, the B-factors are perhaps a little higher around the binding site for one of the two molecules without substrate. Is it possible that substrate was partially occupied in that site? Please provide simulated annealing omit maps for all three binding sites.

In our revised manuscript we have carefully re‐analyzed our data processing and refinement of the Synj1 structure in complex with diC8‐PI(3,4,5)P_3_, suggesting an even higher resolution for this substrate-bound structure of 2.73 Å. Subsequently, we carefully analyzed the unbiased electron density in the active site of the three protein molecules in the asymmetric unit. The omit map for molecule E very clearly shows electron density for diC8‐PI(3,4,5)P_3_, including the phosphate groups on position 3, 4 and 5, allowing us to unambiguously model this compound (see unbiased omit map and 2Fo‐Fc and residual Fo‐Fc maps after refinement in Author response image 1). The omit maps for the other two active sites (chain A and chain C) show some continuous density which might account for the presence of a ligand at low occupancy. The latter is especially true for chain C (Author response image 1). These differences in occupancy could reflect differences in kinetics of access of substrate or release of products from the three active sites, depending on their local environment in the crystal packing, and/or could reflect the substrate being hydrolyzed to different degrees. Since the density in chain A and C is very weak and to avoid overinterpretation, we opted not to model these densities as substrate molecules.

The modelling of the densities in the different active sites is now also being discussed in more detail in the Results section of our revised manuscript.

**Author response image 1. respfig1:** Electron density in the active sites of protein molecules corresponding to chain E (top), C (middle) and A (bottom) of the Synj1_528‐873_ ‐ diC8‐PI(3,4,5)P_3_ structure. The panels on the left show the Fo‐Fc and 2Fo‐Fc map contoured at 3σ and 1σ respectively, while the panels on the right show the omit map contoured at 2.5σ.

A cis-peptide bond is found between Y784 and K785, as in all other 5-phosphatase structures except in the deposited structure of *S. pombe* synaptojanin. It might be instructive to try to fit a cis peptide bond against the *S. pombe* diffraction data.

As suggested by the reviewer, a *cis*‐peptide bond was introduced between the corresponding Y794 and K795 residues of the SPSynj structure (PDB 1I9Z), after which refinement against the diffraction data was performed and electron densities were compared. Author response image 2 shows a comparison of the resulting 2Fo‐Fc map at 1σ and the Fo‐Fc map at 3σ for a model with a *trans*‐peptide bond (Author response image 2) versus a *cis*‐peptide bond (Author response image 2). The resulting maps show a clear preference for the *cis*‐peptide, suggesting that the *cis*‐peptide between Y784 and K785 (or Y794 and K795 in SPSynj) is conserved in all 5PPases. We now also mention the possibility for the occurrence of a cis‐peptide bond in SPSynj in our revised manuscript.

**Author response image 2. respfig2:** Electron density map around residues Y794 and K795 of *S. pombe* Synj (SPSynj). 2Fo‐Fc map at 1σ and Fo‐Fc map at 3σ around residues Y794 and K795 of SPSynj with (A) a *trans‐*peptide bond and (B) a *cis‐*peptide bond.

As the authors point out, the effect of the Y793C mutant might be indirect by affecting nearby residues at the active site. It is particularly interesting that Y793 is close to the cis-peptide bond between Y784 and K785. Is it possible that the mutation destabilizes this cis-peptide bond, corroborating its functional importance?

The Y793 residue is indeed located on the large P4IM loop, in between active site residues Y784‐K785 and K798‐R800. Since the Y793C mutant has a rather mild but broad effect on the catalytic activity for all tested substrates, we anticipate a scenario where this mutation has a subtle effect on the conformation of the entire P4IM loop. At this point it is therefore very hard to uncouple the potential effect of this mutation on the *cis*‐peptide bond from the effect on the other active site residues resulting from changes in loop conformation.

Reviewer #2:Paesmans and colleagues describe the nanobody-bound crystal structure of the 5-phosphatase domain of human Synj1, an important regulator of clathrin-mediated endocytosis, with and without a trapped substrate. Based on the structure, they kinetically characterize catalysis towards different substrates and come up with two alternative mechanisms how Syn1 mediates catalysis. They finally characterize three disease mutants of Syn1, which show reduced activity or aggregate.The paper is well drafted, the nanobody-aided structure determination of Syn1 is elegant and the kinetic experiments appear overall sound. A question to discuss is whether the novelty of the structural data is sufficient for publication in eLife. The structure of yeast synaptojanin is known for a long time and it seems rather similar to the human counterpart (although this is difficult to judge from this manuscript). Furthermore, many features of substrate binding and catalysis were already deduced from the product-bound yeast synaptojanin structure. I have also some questions/doubts about the substrate-bound structure.

With regard to the novelty of the data presented in our manuscript we would like to emphasize that this is the first structure of the human Synj1 protein. Considering the importance of this protein as a potential target for drug design, a structure of the real human protein rather than a homologue will be crucial for any structure‐based drug design effort. Also, note that we show in our study that the substrate specificity profile of the yeast homologue, SPSynj, differs from that of the human Synj1, which shows that SPSynj is not an optimal model system, again emphasizing the importance of a structure from the human protein.

We also provide the very first structure of any 5‐phosphatase protein in complex with the real substrate. In the revised document we carefully re‐processed and reanalyzed the diffraction data, suggesting a higher resolution for the substrate‐bound structure of 2.73 Å, and the PDB validation report shows the good quality of the structure compared to structures of similar resolution. Furthermore, the only objective criterion to address the question of data quality in our believe is to look at the electron density around the bound ligand. We have made a figure showing the electron density around the ligand in our structure and that of SPSynj in complex with the product inositol‐(1,4)bisphosphate (Author response image 3). For comparison we provide the 2Fo‐Fc and Fo‐Fc maps around the ligands of the respective structures as well as the omit maps. Both maps were calculated in exactly the same way for straightforward comparison. I hope that the reviewer will appreciate that (i) the position of the diC8‐PI(3,4,5)P_3_ substrate in our structure is very clearly and unambiguously defined and also the scissile 5‐phosphate group can be modelled in an unambiguous way without the danger of any overfitting; (ii) the ligand in the SPSynj structure is clearly much less well defined in the density and modelling seems rather ambiguous. As discussed in the revised document, careful analysis of the electron density in our re‐processed structure of human Synj1 in complex with diC8‐PI(3,4,5)P_3_ now also reveals density that could account for the nucleophilic water molecule, adding to the information with regard to the mechanism.

Finally, we also performed a very thorough kinetic analysis, showing for the first time the direct contribution of the 4‐phosphate group to catalysis rather than substrate binding. Together with the structural information, this allowed us to propose two potential mechanisms for this enzyme. Additionally, we also performed a first thorough kinetic characterization of disease‐associated mutations showing a clear link between the effect on catalysis and the severity of disease.

**Author response image 3. respfig3:** Electron density around diC8‐PI(3,4,5)P_3_ in the Synj1_528‐873_ structure (PDB 7A17) and around inositol‐(1,4)bisphosphate in the SPSynj structure (1I9Z). The panels on the left show the Fo‐Fc and 2Fo‐Fc map contoured at 3σ and 1σ respectively, while the panels on the right show the omit map contoured at 3σ.

1) Detailed structural comparisons of Syn1 to other 5'-phosphatases (including rmsd) are missing to judge similarities and recognize potential differences between the structures. A structural comparison of the catalytic centers of some 5' phosphatases, including human and yeast synaptojanin with bound ligands, complementing Supplementary file 1, would allow the reader to appreciate (potentially) new mechanistic insights from the Syn1 structure.

An additional supplementary figure (Figure 2—figure supplement 3) has been added to the revised manuscript, showing the structural superposition of our structure with the 5PPase‐domain of INPP5B, SHIP2, OCRL, INPP5E and SPSynj. Also, an extra supplementary table (Supplementary file 1) is added to the manuscript, displaying rmsd values for structural superposition and sequence identities between the relevant 5PPases.

2) A major point of the paper is the substrate bound to Syn1. However, it remains unclear why the substrate has not been turned over after an overnight incubation at 20 C. The authors speculate about a missing catalytic water, but this cannot explain missing catalysis since the substrate-binding site is apparently accessible, in particular for water. Is the crystallization condition incompatible with catalysis? Does nanobody-binding reduce catalysis? Is there at all any substrate left after overnight incubation with the crystals? Without answering this, it remains uncertain whether the observed ligand is indeed a non-converted substrate molecule (see also next point).

We understand the concern raised by the reviewer that a substrate molecule would still be present in the active site after overnight incubation. This question is in part triggered by a lack of detail in the description of the cryo‐solution in our initial manuscript. In contrast to the apo‐structure, for the diC8PI(3,4,5)P_3_‐bound structure the cryo‐solution was also supplemented with 1 mM of the substrate diC8PI(3,4,5)P_3_. As we now explain in detail in our revised manuscript, crystals were transferred to the cryosolution for approximately 30 seconds prior to flash‐freezing in liquid nitrogen. So, it is highly likely that during short soaking in the cryo‐liquid a new substrate molecule re‐entered the active site of one of the protein molecules and was then trapped by flash freezing. This substrate is present with full occupancy and has very clearly determined density, as shown in the refined and omit density maps (see Author response image 1). These issues are now clarified and discussed in the revised document. In addition, the orientation of the substrate in the active site is clearly physiologically relevant as witnessed by the orientation of catalytically important residues vis‐à‐vis the substrate and the high correlation between the structure and our mutagenesis/kinetics results.

As suggested by the reviewer, we also addressed the question on whether either the presence of the nanobody and/or the crystallographic condition (pH) could affect catalysis. First, using the same method as described in the Materials and methods section of the manuscript, we compared the Michaelis‐Menten kinetics for the substrate diC8‐PI(3,4,5)P_3_ for Synj1_528‐837_ in either absence or presence of an excess of Nb15 (Author response image 4). This analysis shows that binding of Nb15 induces a small decrease in the overall catalytic efficiency (k_cat_/K_M_) of the enzyme by a factor of 3.5, potentially by affecting the enzyme dynamics. Secondly, we tested the effect of the buffer and pH of the mother liquor used to crystallize the protein, by comparing enzyme catalysis in HEPES buffer with pH 7.5 and citrate buffer with pH 5.5. This analysis shows a decrease in catalytic efficiency by a factor of 12 at the pH used to crystallize the protein. Cumulative this shows that substrate turnover is slowed down under the conditions used to crystallize the protein, which could have aided in trapping the substrate in a non‐hydrolyzed form. However, this also shows that the enzyme is still catalytically competent under the conditions used, thus proving the relevance of our structure.

**Author response image 4. respfig4:** The 5‐phosphatase activity of Synj1_528‐873_ is affected by Nb15 and acidic pH. (A) Michaelis‐Menten curves obtained by using the Malachite green assay with 2.5 nM of human Synj1_528‐873_ and various concentrations of diC8‐PI(3,4,5)P_3_ in absence or presence of 100 nM Nb15. (B) Michaelis‐Menten curves obtained by using the Malachite green assay with 2.5 nM of human Synj1_528‐873_ and various concentration of diC8‐PI(3,4,5)P_3_ in assay buffer containing 25 mM HEPES at pH 7.5 or in assay buffer containing 25 mM sodium citrate at pH 5.5.

3) Crystallographic data of the substrate-bound crystals: Why is the signal to noise value so low for the substrate-bound crystals? Why did the authors apply anisotropic corrections to their data? Did the crystals diffracted anisotropically – if yes, provide some more information, e.g. to which resolution along a*,b*,c*. Judging from the spherical completeness, the dataset rather correspond to a 3.x A resolution dataset in terms of reflection numbers.

In the revised document we carefully re‐processed and re‐analyzed the diffraction data for the crystal of the substrate‐bound protein. This analysis suggested an even higher resolution for the substratebound structure of 2.73 Å. These resolution limits were automatically set using the autoPROC pipeline (https://www.globalphasing.com/autoproc/manual/autoPROC4.html), using a diffraction cut‐off of I/sigI = 1.4. Since the anisotropy analysis by STARANISO indeed indicated anisotropic diffraction, the anisotropy correcteddata from STARANISO was used, which was automatically generated in the autoPROC pipeline. The anisotropic diffraction is shown in Appendix 1—figure 1, which was generated by STARANISO.

The diffraction limits along the axes of the ellipsoid fitted to the diffraction cut‐off surface used by STARANISO for the Synj1_528‐873_‐diC8‐PI(3,4,5)P_3_ data are:

2.86 Å along 0.043 a* + 0.999 c*

2.72 Å along b*

3.14 Å along ‐0.974 a* + 0.228 c*

All details about the processing and refinement strategy and the diffraction limits along the axes of the ellipsoid are now provided in the Materials and methods section of the revised document and Appendix 1—figure 1, which shows this information for both structures.

The application of the anisotropy correction by STARANISO provided an outstanding improvement of the data statistics as well as the map quality. Author response table 1 shows that the dataset would correspond to approximately 3.0 Å resolution without anisotropy correction, based on CC(1/2) and on I/sigI = 1.4 cut‐off criteria. Anisotropic correction improved this to 2.73 Å resolution.

**Author response table 1. resptable1:** 

	diC8‐PI(3,4,5)P_3_‐bound Synj1_528‐873_
	STARANISO 2.73 Å	Without STARANISO 2.73 Å	Without STARANISO 3.00 Å
PDB number	7A17		
**Data collection**			
Resolution range (Å)^a^	87.39‐2.73 (3.02‐2.73)	87.39‐2.73 (2.83‐2.73)	87.39‐3.00 (3.16‐3.00)
Spherical completeness (%)^a^	76.2 (22.8)	99.8 (100)	99.8 (99.9)
Ellipsoidal completeness (%)^a^	91.5 (57.3)	/	/
Unique reflections	32149	42013	31720
Mean (I)/SD(I)^a^	5.3 (1.4)	4.1 (0.7)	5.2 (1.4)
CC(1/2)^a^	0.964 (0.474)	0.953 (0.195)	0.962 (0.448)
Multiplicity^a^	3.5 (3.6)	3.5 (3.6)	3.5 (3.4)
Rmeas (%)a	28.2 (123.7)	37.1 (259.9)	28.1 (114.4)

^a^ Values in parentheses are for the high‐resolution shell.

While resolution limits are by default somewhat arbitrary and the optimal cut‐off criteria (if any at all) are being debated in the field, the real criterion for judging the quality of data and for comparing different processing strategies lies in the final density. As can be judged from Author response image 5, the level of detail in the unbiased omit map around the ligand improved after using STARANISO and the quality of this unbiased omit map allowed to place the substrate with high confidence in the active site.

**Author response image 5. respfig5:** Omit map contoured at 3σ around the diC8‐PI(3,4,5)P_3_ substrate of a structure (A) processed with STARANISO to 2.73 Å, (B) processed without STARANISSO to 2.73 Å and (C) processed without STARANISO to 3.0 Å. Panel (D) Refined 2Fo‐Fc map contoured at 1σ around the diC8‐PI(3,4,5)P_3_ substrate of a structure processed with STARANISO to 2.73 Å.

It is astonishing that for such low resolution, 91 water molecules with an average B-factor of only 34 Angstroms could be modelled (I would hardly expect to see any visible water molecules at this resolution). Is the model overfitted? Any explanation for the wide gap of Rwork and Rfree? What was the exact refinement strategy? Overall, the data quality of the substrate-bound crystals is borderline to accurately evaluate ligand binding, in particular if the identity of the bound ligand is not 100% clear.

After re‐analysis (as outlined above) and re‐refinement of the data for the Synj1_528‐873_‐diC8‐PI(3,4,5)P_3_ structure, the obtained resolution was set at 2.73 Å. The refinement strategy consisted of iterative cycles of manual building and refinement with Phenix.Refine. Initial rounds of the refinement strategy included XYZ coordinate, real‐space and individual B‐factors refinement, and towards the final cycles optimization of the X‐ray/stereochemistry weights was included. Finally, the structure was submitted to the PDB‐REDO server (Joosten et al., 2014) for a final optimization. After, critical assessment of all the water molecules, some water molecules were deleted which led to a final amount of 71. This seems a very reasonable amount of water molecules considering the resolution and the large amount of amino acid residues in the asymmetric unit (3 Synj1_528‐873_ chains and 3 Nanobody chains). For a comparison of expected water molecules per amino acid residues at different resolution limits we could e.g. refer to Carugo, Oliviero, and Domenico Bordo. "How many water molecules can be detected by protein crystallography?." Acta Crystallographica Section D: Biological Crystallography 55, no. 2 (1999): 479‐483.

In the final refined Synj1_528‐873_‐diC8‐PI(3,4,5)P_3_ structure the R and R_free_ values are 19.88 % and 25.74%, respectively, corresponding to a R‐R_free_ gap of 5.86%. As can be judged from statistics of R, R_free_ and RR_free_ gaps from structures of similar resolution (Author response table 2, as given by *phenix.r_factor_statistics*), this gap is perfectly expected for a good structure at a resolution of 2.73 Å resolution.

**Author response table 2. resptable2:** R, R_free_ and R‐R_free_ statistics for structures in the resolution range 2.50‐2.90 Å as given by phenix.r_factor_statistics.

Histogram of R_work_ for models in PDB at resolution 2.50‐2.90 Å : 0.119 ‐ 0.171 : 281 structures 0.171 ‐ 0.223 : 5171 structures 0.223 ‐ 0.274 : 3263 structures 0.274 ‐ 0.326 : 122 structures 0.326 ‐ 0.378 : 4 structures
Histogram of R_free_ for models in PDB at resolution 2.50‐2.90 Å: 0.159 ‐ 0.220 : 469 structures 0.220 ‐ 0.281 : 6011 structures 0.281 ‐ 0.343 : 2315 structures 0.343 ‐ 0.404 : 45 structures 0.404 ‐ 0.465 : 1 structures
Histogram of R_free_‐R_work_ for all model in PDB at resolution 2.50‐2.90 Å: 0.001 ‐ 0.021 : 408 structures 0.021 ‐ 0.041 : 2370 structures 0.041 ‐ 0.060 : 3841 structures 0.060 ‐ 0.080 : 1809 structures 0.080 ‐ 0.100 : 413 structures
Number of structures considered: 8841

We would also like to point out that the electron density maps in the final structures look very convincing (see e.g. Author response image 6 for electron density in representative parts of the structure) and allowed to unambiguously model the substrate diC8‐PI(3,4,5)P_3_ as shown in Author response image 1, Author response image 3 and Author response image 5.

Finally, we also attached the PDB validation reports with this re‐submission, showing the good quality of our structures in comparison to other structures of similar resolution.

**Author response image 6. respfig6:** 2Fo‐Fc map contoured at 1 **σ** around (A) residues 670‐678 and (B) residues 723‐730 of chain A of the Synj1_528‐873_diC8‐PI(3,4,5)P_3_ structure.

Reviewer #3:The authors report crystal structures of the apo and substrate bound versions of the inositol 5-phosphatase domain of human synaptojanin-1 (Synj1), crystallized in the presence of a nanobody. The overall structure is similar to those of several previously solved 5-phosphatase enyzmes from yeast and humans. The main addition to the field in the current structure with respect to past work is that a substrate is present in this case, whereas previous structures were apo or product-bound.

We thank the reviewer for his assessment and comments.

For the comment with regard to the novelty of our findings we would make the following remarks (see also our answer to reviewer 2):

This is the first structure of the human Synj1 protein. Considering the importance of this protein as a potential target for drug design, a structure of the real human protein rather than a homologue will be crucial for any structure‐based drug design effort. Also, note that we show in our study that the substrate specificity profile of the yeast homologue, SPSynj, differs from that of the human Synj1, which shows that SPSynj is not an optimal model system again emphasizing the importance of a structure from the human protein.

We also provide the very first structure of any 5‐phosphatase protein in complex with the real substrate. In the revised document we carefully re‐processed and re‐analyzed the diffraction data, suggesting a higher resolution for the substrate‐bound structure of 2.73 Å. As discussed in the revised document, careful analysis of the electron density in our re‐processed structure of human Synj1 in complex with diC8‐PI(3,4,5)P_3_ now also reveals density that could account for the nucleophilic water molecule, adding to the information with regard to the mechanism.

Finally, we also performed a very thorough kinetic analysis, showing for the first time the direct contribution of the 4‐phosphate group to catalysis rather than substrate binding. Together with the structural information, this allowed us to propose two potential mechanisms for this enzyme. Additionally, we also performed a first thorough kinetic characterization of disease‐associated mutations showing a clear link between the effect on catalysis and the severity of disease.

The structure contains 3 catalytic chains per asymmetric unit. The ligand-bound form was obtained by soaking apo crystals in a di-C8 form of the substrate phosphoinositide. The structure of the complex was determined at 2.9 Å resolution. Detail on the catalytic geometry is limited at this resolution. Water and metal ion positions are not visualized at all, or with limited accuracy.

As explained above, in our revised manuscript we have carefully re‐analyzed our data processing and refinement of the Synj1 structure in complex with diC8‐PI(3,4,5)P_3_ suggesting an even higher resolution for this substrate‐bound structure of 2.73 Å. Subsequently, we carefully analyzed the unbiased electron density in the active site of the three protein molecules in the asymmetric unit. The omit map for molecule E very clearly shows electron density for diC8‐PI(3,4,5)P_3_, including the phosphate groups on position 3, 4 and 5, allowing us to unambiguously model this compound (see unbiased omit map and 2Fo‐Fc and residual Fo‐Fc maps after refinement in Author response image 1). The orientation of the substrate in the active site is clearly physiologically relevant as witnessed by the orientation of catalytically important residues vis‐à‐vis the substrate and the high correlation between the structure and our mutagenesis/kinetics results. The excellent quality of the density in the active site of chain E also revealed additional electron density that could be accounted for by the nucleophilic water molecule. We attached the PDB validation reports with this re‐submission, showing the good quality of our structures in comparison to other structures of similar resolution.

Only one of the three catalytic chains binds to substrate. I did not see an explanation for the lack of binding to the other two chains. The 5-phosphate is still present on the substrate, despite that catalytically active wild-type enzyme was used. The authors present a detailed scheme for the catalytic mechanism on the basis of their structure, but I did not see an explanation for the failure of the enzyme to hydrolyze the 5-phosphate. I would speculate that the pH 5.0 of the crystallization could account for this. Have the authors attempted to adapt the crystals to neutral pH? Is a non-hydrolyzable substrate analogue available, or could it be synthesized? (I appreciate that phosphoinositide synthetic chemistry is non-trivial.) It would certainly be helpful for the field to know the geometry of a catalytically productive substrate complex, yet I am not convinced that the authors have as yet succeeded in obtaining such a structure. In my opinion, the conclusions concerning enzyme mechanism are, at this stage, over-interpreted.

We understand the concern raised by the reviewer that a substrate molecule would still be present in the active site after overnight incubation. This question is in part triggered by a lack of detail in the description of the cryo‐solution in our initial manuscript. In contrast to the apo‐structure, for the diC8PI(3,4,5)P_3_ structure the cryo‐solution was also supplemented with 1 mM of the substrate diC8PI(3,4,5)P_3_. As we now explain in detail in our revised manuscript, crystals were transferred to the cryosolution for approximately 30 seconds prior to flash‐freezing in liquid nitrogen. So, it is highly likely that during short soaking in the cryo‐liquid a new substrate molecule re‐entered the active site of one of the protein molecules and was then trapped by flash freezing. This substrate is present with full occupancy and has very clearly determined density, as shown in the refined and omit density maps (see Author response image 1, included in this rebuttal letter). These issues are now clarified and discussed in the revised document.

As suggested by reviewers 2 and 3 we also investigated the influence of the Nb and the pH of the crystallization conditions on Synj1 catalytic activity. As detailed in our answer to reviewer 2 (see Author response image 4), we find that the presence of the Nb induces a small decrease in the overall catalytic efficiency (k_cat_/K_M_) of the enzyme by a factor of 3.5, potentially by affecting the enzyme dynamics, while a decrease of pH from 7.5 to 5.5 causes a decrease in catalytic efficiency by a factor of 12. Cumulative this shows that substrate turnover is slowed down under the conditions used to crystallize the protein, which could have aided in trapping the substrate in a non‐hydrolyzed form. However, this also shows that the enzyme is still catalytically competent under the conditions used, thus showing the relevance of our structure.

Concerning the presence of clear density for the non‐hydrolyzed substrate in only one of the active sites, as stated above, it is true that the other two active sites (mainly chain C) show only some continuous density which might account for the presence of a ligand at low occupancy (see Author response image 1). These differences in occupancy could reflect differences in kinetics of access of the substrate or release of products from the three active sites, depending on their local environment in the crystal packing, and/or could reflect the substrate being hydrolyzed to different degrees.

This being said, we think it is not unusual that ligands are only found occupying a subset of the binding pockets present in the asymmetric unit, despite being seemingly accessible via solvent channels. A survey at the CCP4 bulletin board and through the literature yielded several similar cases, some of which we mention here:

– Structure of penicillin‐binding protein 2A in complex with a quinazolinone (PDB 6Q9N), where only one of the two protein chains in the active site contained density for the compound (Janardhanan *et al.*, 2019. DOI: 10.1128/AAC.02637‐18).

– Structure of the N‐terminus of a glycosidase from *Cellulosimicrobium cellulans* in complex with 1deoxymannojirimycin (DMNJ) (PDB 4AQ0), where only one of the two chains in the asymmetric unit contained the ligand DMNJ, while the other contained a solvent molecule (Bis‐Tris propane) (Tiels *et al.*, 2012. DOI: 10.1038/nbt.2427).

– Structure of NADP^+^‐specific isocitrate dehydrogenase from *Corynebacterium glutamicum* (PDB 3MBC) where the co‐enzyme was present in only one of the two biologically monomeric NCS‐related protein molecules in the asymmetric unit (Sidhu *et al.,* 2011. DOI: 10.1107/S0907444911028575).

– Structure of holo‐transaminase soaked with an acceptor substrate (α‐ketoglutarate) (PDB 5E25), with nice ligand density observed in only one of the three monomers in the asymmetric unit (currently not published).

– Multiple examples in a paper reporting the *in crystallo‐*screening for discovery of human norovirus 3C‐like protease inhibitors (Guo et al. J Struct Biol. 2020 4:10003. DOI: 10.1016/j.yjsbx.2020.100031), e.g. PDB 6T49, 6T6W, 6T2I.

The paper is, apart from the omissions noted above, very thorough. It reads like a review article in the Introduction and Discussion. The results includes more detail than is standard. Overall the paper would be more readable if it were edited to about half of the current length.

As suggested by reviewer 2 and 3, we significantly shortened the manuscript in certain parts, taking care not to lose information.